# Evaluating the Suitability of Urban Expansion Based on the Logic Minimum Cumulative Resistance Model: A Case Study from Leshan, China

**Haijun Wang** [1,2,*] , **Peihao Peng** [1] , **Xiangdong Kong** [2,3] , **Tingbin Zhang** [1,2] and **Guihua Yi** [1]

1   College of Earth Science, Chengdu University of Technology, Chengdu 610059, China
2   Engineering and Technical College of Chengdu University of Technology, Leshan 614000, China
3   School of Civil Engineering and Architecture, Southwest Petroleum University, Chengdu 610500, China
*   Correspondence: wanghaibo.2006@163.com; Tel.: +86-17369142382

**Abstract:** This paper focuses on the suitability of urban expansion in mountain areas against the background of accelerated urban development. Urbanization is accompanied by conflict and intense transformations of various landscapes, and is accompanied by social, economic, and ecological impacts. Evaluating the suitability of urban expansion (UE) and determining an appropriate scale is vital to solving urban environmental issues and realizing sustainable urban development. In mountain areas, the natural and social environments are different from those in the plains; the former is characterized by fragile ecology and proneness to geological disasters. Therefore, when evaluating the expansion of a mountain city, more factors need to be considered. Moreover, we need to follow the principle of harmony between nature and society according to the characteristics of mountain cities. Thus, when we evaluate the expansion of a mountain city, the key procedure is to establish a scientific evaluation system and explore the relationship between each evaluation factor and the urban expansion process. Taking Leshan (LS), China—a typical mountain city in the upper Yangtze River which has undergone rapid growth—as a case study, the logic minimum cumulative resistance (LMCR) model was applied to evaluate the suitability of UE and to simulate its direction and scale. The results revealed that: An evaluation system of resistance factors (ESRFs) was established according to the principle of natural and social harmony; the logic resistance surface (LRS) scientifically integrated multiple resistance factors based on the ESRF and a logic regression analysis. LRS objectively and effectively reflected the contribution and impact of each resistance factor to urban expansion. We found that landscape, geological hazards and GDP have had a great impact on urban expansion in LS. The expansion space of the mountain city is limited; the area of suitable expansion is only 23.5%, while the area which is unsuitable for expansion is 39.3%. In addition, it was found that setting up ecological barriers is an effective way to control unreasonable urban expansion in mountain cities. There is an obvious scale (grid size) effect in the evaluation of urban expansion in mountain cities; an evaluation of the suitable scale yielded the result of 90 m × 90 m. On this scale, taking the central district as the center, the urban expansion process will extend to the neighboring towns of Mianzhu, Suji, Juzi and Mouzi. Urban expansion should be controlled in terms of scale, especially in mountain cities. The most suitable urban size of LS is 132 km$^2$. This would allow for high connectivity of urban-rural areas with the occupation of relatively few green spaces.

**Keywords:** mountain city; urban expansion; resistance evaluation system; Logic minimum cumulative resistance model

## 1. Introduction

Over the past two decades, there has been increasing spatial expansion from green spaces (e.g., parks, woodlands, and grasslands) to constructed land (e.g., residential and industrial land), especially in mountain cities in Asia which face severe urban population pressure [1–3]. The spatial expansion of mountain cities has increasingly been a key characteristic of land use change, and the quantity of built-up areas is increasing at a rate of 5.5% per year [4]. The conversion from green spaces to constructed land represents the most significant change in the process of urbanization [5,6]. Therefore, coordinating the protection of green spaces and urban spatial expansions to maximize the use of constructed land and realize sustainable development is a key challenge associated with urban expansion [7]. Studies on UE (urban expansion) generally focus on the expansion of constructed land; cities are complex social and natural ecosystems [8]. UE has caused serious economic, social, and environmental issues in the southwest mountain cities of China. However, relatively few studies have evaluated the consequences of spatial expansion of such cities according to the characteristics of mountain cities, such as limited land sources, frequent geological hazards and ecological sensitivity. The urbanization rate in China reached 59.58% in 2018 [9]. The average annual growth rate of urbanization in Sichuan province has been 7.21% higher than the national average over the past 15 years. Rapid urbanization has not only caused the area of green spaces and farmlands to decrease dramatically, but also threatens ecological security [10–13]. Subsequently, how to scientifically control UE and coordinate urbanization and ecological security against the background of rapid expansion has become a research focus.

A mountain city in Southwest China, LS has gradually entered the "big traffic, big business, big cities" era, with the development of high-speed railway infrastructure, an airport, and a highway. As urbanization has significantly sped up (see Figure 1) and the social economy has rapidly developed over the past 15 years, UE has caused a reduction in the amount of green spaces and an increase in the occupation of arable land. Therefore, determining the best ways of reducing stress on urban ecology, determining the direction of expansion, and reasonably transforming non-constructed land into constructed land are inevitable problems. In the past, many researchers have used a spatial dynamic model and an empirical model to evaluate the suitability of UE. The spatial dynamic model explores the relationship between UE and its impact factors, and then simulates such expansion using some algorithms, such as Logic-Cellular Automaton Markov (LCAM) [14,15], generalized regression neural network (GRNN) [16,17], and the Sleuth model [18,19]. The evaluation process of the empirical model may be divided into two steps. First, the weight of factors is allocated by experience or expert scoring. Second, the weights are used in a spatial analysis method to perform an evaluation; such methods include the ordered weighted averaged (OWA) and the analytic hierarchy process (AHP) [20–22]. Both can be used to evaluate UE, but each has its drawbacks, e.g., being overly dependent on experiences, or the need to set complex parameters.

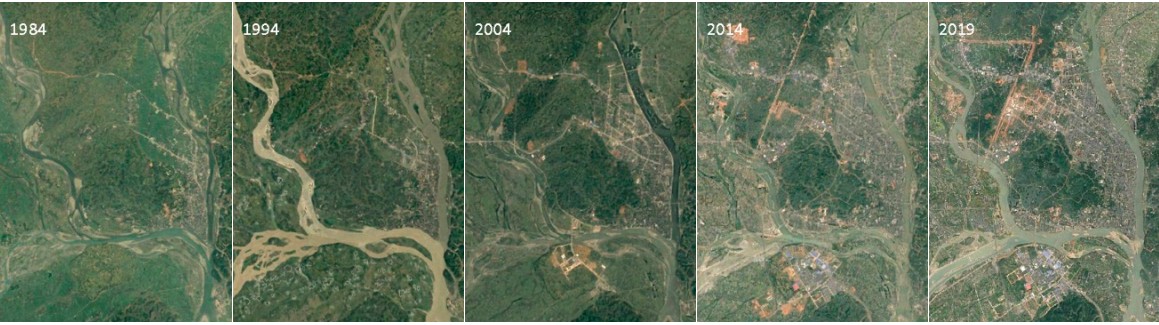

**Figure 1.** Urban expansion from 1984 to 2019 in LS (Google Maps).

Mountain cities have more complex environments than other cities. For example, they have irregular terrain, limited available land resources, dense river networks and a high incidence of geological disasters. More field investigations, yielding more measured parameters, are necessary

to evaluate UE for spatial dynamic models (e.g., LCAM, GRNN and SLEUTH). Empirical model (e.g., OWA and AHP) depend more on the judgment and the experience of one or more experts. However, in applications involving complex mountain areas, some comprehensive expert experience is not available, and therefore the model simulation result is not objective. Therefore, we need to find a new method or improve the existing methods to better evaluate the expansion of mountain cities. The minimum cumulative resistance (MCR) model originated from the study of the diffusion processes of species [23]. The MCR model is regarded as an effective method [24–29] for evaluating landscape connectivity because of the speed of its algorithms, the simple data structure, and the visual results. It has been used to evaluate the urban ecological security and urban ecological planning [18]. The model is based on ecology principles; at its core are the competitive and conflicting processes between the source of UE and other landscape units. The source must overcome resistance from the other landscape units, such as rivers, woodland, and farmland. The advantage of this model is that it considers both the influence of resistance factors and the distance to the source in the process of UE.

However, there are still some limitations and problems for this model in the evaluation of urban expansion in mountain areas. To solve these problems and better evaluate the urban expansion of mountain cities, we need to improve the MCR model. Therefore, by taking LS as the case study, we use the improved MCR model (LMCR) to evaluate the suitability of expansion. This case study provided a vital contribution to reconciling urban expansion with the protection of urban ecology in human settlements. More specifically, this study attempted to address the following questions:

(1)    What were the impact factors related to urban expansion in LS?
(2)    How suitable was the urban expansion?
(3)    What was the appropriate direction for UE and how were the urban rural areas connected under a specific scale scenario?

## 2. Materials and Methods

### 2.1. Study Area

Leshan, a third-tier city in China, located in the southwest of the Sichuan basin and upper Yangtze River (103.3°–104.1°E, 29.0°–30.0°N). It is 120 km away from Chengdu (the capital city of the Sichuan province) and 270 km from Chongqing (one of four municipalities under the direct administration of the central government). It is an important part of the C-Y (Chengdu and Chongqing) urban agglomeration (Figure 2), and it lies in the southwest mountain region in the Sichuan Basin. The southwest topography is higher than that of the northeast (Figure 3). Furthermore, it is a famous ecotourism city with pleasant weather and numerous historical heritage sites. By the end of 2018, LS had four districts, seven counties, and one county-level city with a total coverage of 847 km$^2$; its population was 1 million. The central district lies at the confluence of the Minjiang River, Dadu River, and Qingyi River. In LS, the terrain is dominated by low mountains, with dense river networks, and limited available land resources. Currently, with the rapid economic development and urban expansion in this area, conflict between constructed land and other landscape types has occurred, causing ecological imbalance.

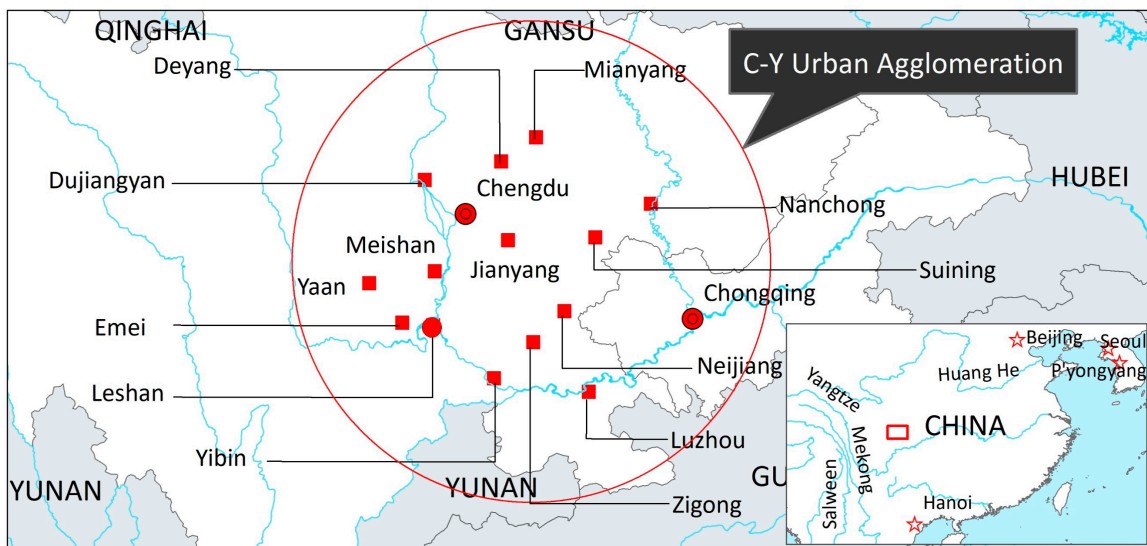

**Figure 2.** The geographical location of Leshan (LS) and its distribution in the Chengdu and Chongqing (C-Y) urban agglomeration.

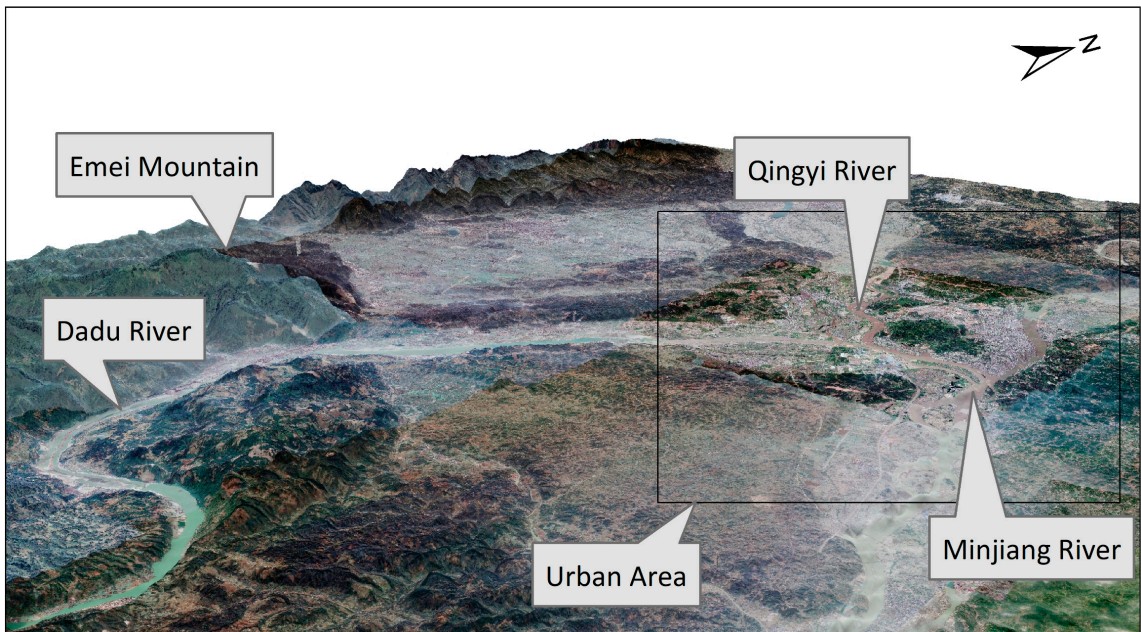

**Figure 3.** Three-dimensional geographic environment of LS and the surrounding area.

### 2.2. Materials

We obtained the land use and land cover (LULC) dataset (1:100,000, 2011) of Sichuan from the Cold and Arid Regions Science Data Center (CASDC). Resource No.3 (ZY03) satellite remote sensing image (July 2018) was downloaded from the China Centre for Resources Satellite Data and Applications (CCRSDA). The image was taken with a multi-spectral camera mounted on the ZY03 satellite. It has four spectral bands with a wavelength range of 0.45–0.89 μm, a spatial resolution of 5.8 m, and a width of 51 km (see Figure 4). Digital elevation data (resolution of 30 m) were from the United States Geological Survey (USGS); statistical data of the urban population and GDP (1995–2018) were provided by the LS bureau of statistics (LSBS). The urban planning and land use thematic maps were provided by the LS land and resources bureau; geological hazards and subgrade bearing capacity data were provided by Sichuan province's No.207 geological survey team.

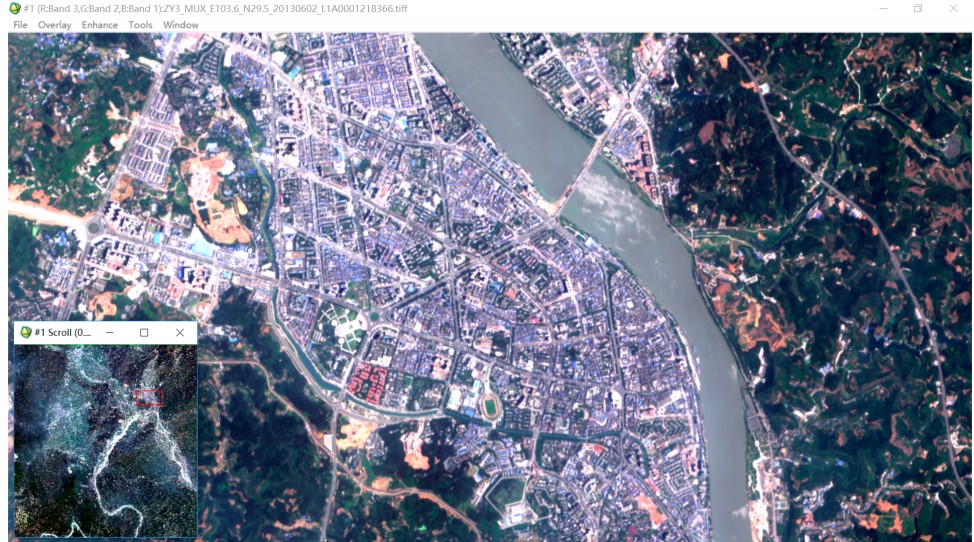

**Figure 4.** True color image of LS taken in June 2018 with a multi-spectral sensor mounted on the ZY03 satellite.

The method of data preprocessing was as follows (see Figure 5). (1) We used the resources satellite No.3 image to update the LULC dataset (2011). Then, we merged the subtypes into more general types; for example, sparse woods were merged into woods. We thus achieved five types of land use and land cover for LS. The five area types were woodland, grassland, farmland, water body, and constructed land. We extracted reservoirs, rivers, roads, built-up areas, pits, and greenbelts from the remote sensing image (ZY03). Vector data were converted to digital raster graphic (DRG) using buffer analysis, e.g., convenient transportation (CT) and the water supply-drainage conditions (WSDC). (3) Statistical and monitoring data, including the density of the urban population (DUP), geological hazards (GH), gross domestic product (GDP), and subgrade bearing capacity (SBC), were interpolated into DRG by the Kriging interpolation [30] method. In view of the size of the study area and the spatial resolution of the raw data (DEM), the output raster resolution was set at 30 m in the buffer analysis and Kriging interpolation. In addition, all of the DRGs were projected into the Universal Transverse Mercator (UTM) projection and World Geodetic System (WGS) 1984 reference system and clipped with an urban bounder. The data preprocessing flow is shown in Figure 5.

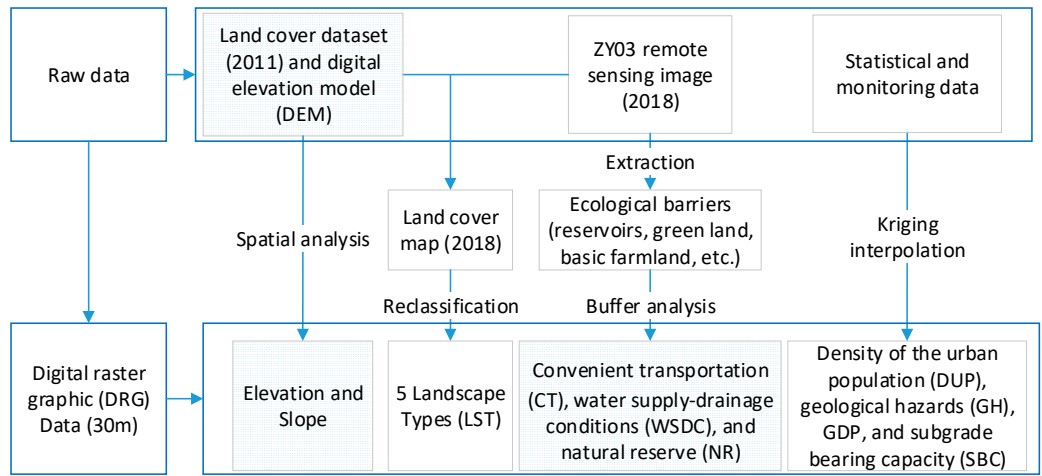

**Figure 5.** Flow chart of data preprocessing.

*2.3. Methods*

LCAM is a coupling model that establishes the relationship between a certain type of land use and land cover (LULC) and the related driving force factors through the logic regression model, and then uses the CA-Markov model to predict land use and cover change (LUCC) [14,15]. GRNN is a memory-based network that can be used for fitting nonlinear relationships. An input layer, pattern layer, summation layer, and output layer are needed to build a GRNN. Thereafter, the weight is calculated based on the nonlinear relationships [16,17]. The SLEUTH model is also performed to simulate the LUCC; its underlying principle resembles that of the CAM model. In detail, the transfer matrix of LUCC in different periods is obtained, then the transfer matrix is used as a rule to simulate the LUCC of a third period (the same step size) based on the different scenarios; the simulation result can be verified using the true land use map of the third period [18,19]. Our analysis showed that spatial dynamic models (LCAM, GRNN, and SLEUTH) performed well in terms of solving the problem of the contribution degree of the driving force factors (impact factors), which is also known as the "weight of the factor". In terms of the weight calculation, spatial dynamic models have more obvious advantages than empirical models (OWA, AHP, etc.), which calculate weight assignment using expert experience and judgment [20–22]. Spatial dynamic models, especially the coupling model, can be used as a vital reference in improving the MCR model. For the improvement of the MCR model, it is key to establish the nonlinear relationship between the resistance factor and the expansion source, and then to obtain the resistance degree of the resistance factor to the expansion source so as to calculate the more objective resistance surface. According to this principle, the MCR model was improved, thus becoming the LMCR model. In this study, we used this new model to evaluate the suitability of urban expansion.

2.3.1. Principle of the LMCR Model

We divided the heterogeneous landscape into "source" or "sink" landscapes according to eco-process theory. A source landscape refers to a landscape that can promote the development of the process, and a sink landscape refers to a landscape that can prevent or delay the development of the process. Whether a landscape is a source or sink landscape is relative: a landscape may be a source in one process, but a sink in another. Therefore, the analysis of source and sink landscapes must be directed at a specific process. The source and sink landscape theory aims to explore the dynamic relationship between urban expansion and ecological protection, which in turn helps us realize a suitable spatial pattern for urban expansion. The transformation between source and sink is a competitive process achieved by overcoming various resistance forces. All of the resistance forces are integrated into a logic resistance surface; in other words, the logic resistance surface (LRS) is used as the cost resistance of source expansion. The LMCR model essentially reflects the minimum cumulative resistance (least-cost) of urban expansion sources to sinks during expansion. Certainly, the least-cost varies with different sinks, and can be achieved using the cost distance analysis based on the cost surface (LRS) and the distance between sources and sinks. In addition, the cost distance in the model is not the actual distance, but the reflection of the spatial relationship between the two landscape units. We achieved the spatial relationship by calculating the drag coefficient when the expansion source passes through other landscape units. We can determine the connectivity between the two units based on the value of the minimum cumulative resistance, when the urban expansion source passes through a specific landscape. Usually, the minimum cumulative resistance path is the suitable path of urban expansion. The principle of the LMCR model is as follows:

$$\text{LMCR} = f_{min}\left(\sum\nolimits_{j=n}^{i=m} D_{ij} \times R_i \times G\right), \tag{1}$$

where LMCR is the logic minimum cumulative resistance; f is a positive correlation function, which reflects the relationship between the minimum resistance and distance from one source (different grades) to another landscape in space and the characteristics of the resistance surface, and *min* denotes the minimum resistance overcame from a source j to a landscape unit i. $D_{ij}$ is the spatial distance

between landscape unit i and source unit j; $R_i$ represents the resistance coefficient that exists in the transition from landscape unit i to source unit j; and $G$ is the level of the source (the urban expansion source can be divided into different levels according to its degree of development).

### 2.3.2. Assumptions of the LMCR Model

According to the principle of the LMCR model and the urban–rural gradient structure [31–33] to implement the model, we need to make four assumptions.

(1)    The urban area is divided into two types by its use: constructed land, which is land that is suitable for urban expansion, and green space (e.g., park, grass land, and garden plots), which is land that is suitable for protection.

(2)    To accelerate economic development and urban construction, constructed land requires expansion. Conversely, to improve the quality of the urban ecosystem, green spaces require expansion. The challenge is to realize a dynamic balance between green space and constructed land.

(3)    Unreasonable urban expansion, in other words, the "sprawl" can be corrected by setting up ecological barriers (such as natural reserves, eco-parks). These control urban sprawl. Moreover, ecological barriers effectively protect the urban ecosystem. That is to say, green spaces could be restrictive factors or driving factors in different processes.

(4)    The urban expansion source is heterogeneous and whether a zone is suitable for urban expansion is identified by the LMCR value.

### 2.3.3. Implementation of the LMCR Model

Some researchers have applied the MCR model to study the process of urbanization [21–24]. Through the analysis of previous research, we found that the model was not perfect and some limitations needed to be improved, especially for the UE of mountain cities. First, the weights allocation of resistance factors was still being conducted by expert scoring. Second, the problem of homogenization of the expansion source was not solved. Viewed from this perspective, the MCR model is inaccurate when evaluating urban expansion in mountain cities. Accordingly, we improved this model and used it to evaluate the urban expansion of LS. The specific flow chart of the improved LMCR model is shown in Figure 6.

(1) Extracting the Expansion Source

The source refers to the type of landscape that promotes the development of urbanization. The source is an area already built up or an area under construction. The source can be divided into urban core areas, urban surrounding areas, and satellite towns according to the urban-rural gradient structure and the intensity of urban land development (ratio of constructed land area to total land area), GDP, and DUP. When performing the LMCR calculation, the input of the source grid can be a patch or a combination of patches, and the source grid can be connected or unconnected in space. The urban expansion source was used in this case study as a source grid, and all of the non-source grids had no value. The expansion source used was extracted and updated from the resources satellite No.3 image (2018) and the LULC dataset (2011). The specific extraction process is described in Section 2.2.

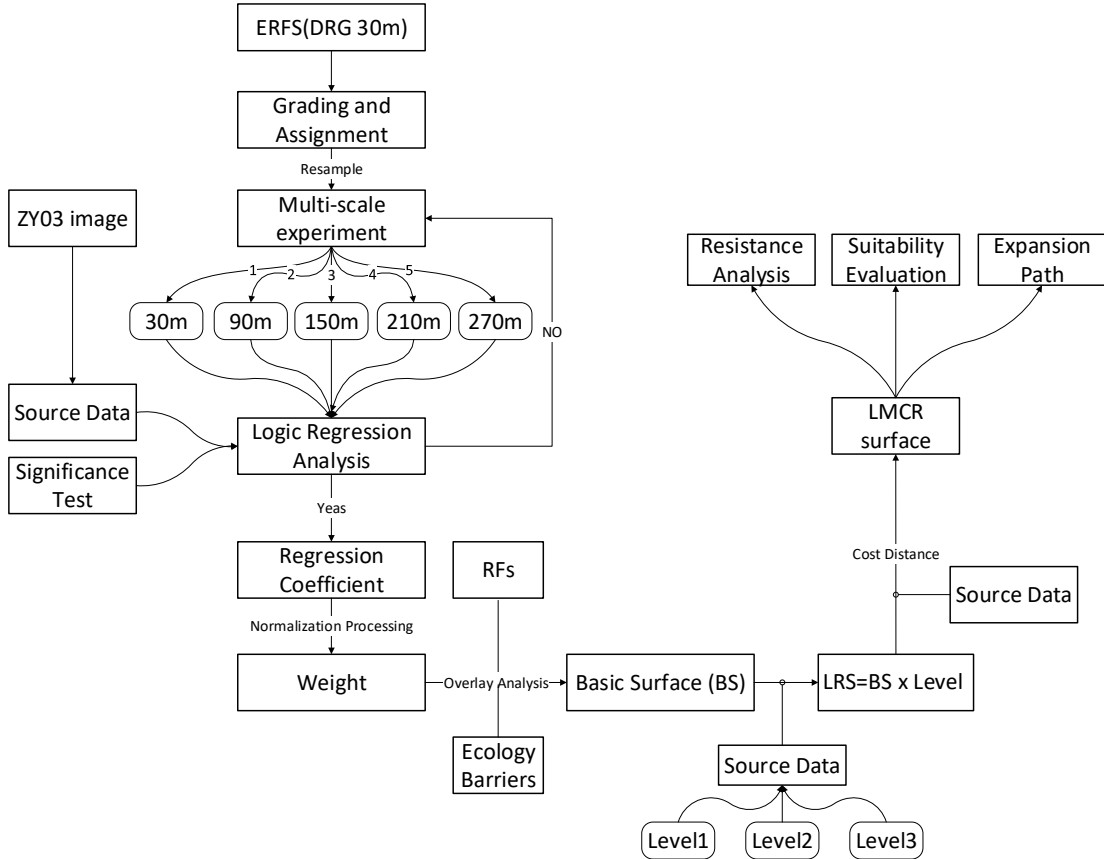

**Figure 6.** Flow chart of the improved Minimum Cumulative Resistance (Logic minimum cumulative resistance) model.

(2) Calculating the LRS

Step 1: Establishing the ESRF (Evaluation System of Resistance Factor)

The selection of resistance factors and the establishment of the ESRF is indispensable for the calculation of the LRS. It is very different for the resistance factors selected in the different cities. Therefore, when selecting resistance factors, we comprehensively investigated and analyzed the natural and social environment of LS. In detail, LS is located in a mountain area with loose soil, a dense river network, and rich precipitation. The special conditions of the area are its proneness to flooding and geological disasters. Accordingly, geological hazards (GH), subgrade bearing capacity (SBC), and water supply-drainage conditions (WSDC) are the important resistance factors affecting urban expansion. In the past 15 years, LS witnessed rapid economic growth and a dramatic increase in urban population, which accelerated urbanization (Figure 1). Therefore, from the perspective of social economy, convenient transportation (CT), the density of the urban population (DUP), and the gross domestic product (GDP) are also important factors affecting LS's development. Moreover, we referred to the urban evaluation and planning standards (UEPS) [25] enacted by the Ministry of Housing and Urban-Rural Development of China (MHURD). Finally, we selected four categories including 10 resistance factors to establish the ESRF. Specifically, these are elevation, slope, GH, SBC, WSDC, CT, DUP, GDP, natural reserve (NR), and landscape type (LST). The 10 resistance factors were divided into two types: one could be overcome (e.g., grassland) and the other could not be overcome (e.g., ecological barriers such as natural reserves, water source areas, and protected basic farmland areas). The selected resistance factors and their ecological analysis are shown in Table 1.

**Table 1.** Selected resistance factors and their ecological analysis (RF represents the resistance factor).

| RF | Ecological Analysis |
|---|---|
| Elevation | LS is located in a hilly area with altitudes ranging from 147 to 641 m. The built-up areas are mainly distributed on relatively flat areas on both sides of the river, and the available construction land is limited. Mountain terrain is the restrictive factor of urban expansion. In view of this, elevation was determined to have an important impact on urban expansion. |
| Slope | The slope of LS ranges from 0° to 70.4°, and the proportion from 14.3° to 27.8° is about 49%. There is a high incidence of landslides and debris flows. Therefore, slope and elevation are combined to evaluate the resistance of urban expansion in this case study. |
| Geological hazards (GH) | According to the survey data, the types of geological hazards in LS include landslides, debris flows, and earthquakes. They mainly occur in the central district, Suji, Juzi, and Mianzhu. Geological hazards are regarded as an important resistance factor for urban expansion (UE). |
| Subgrade bearing capacity (SBC) | The urban expansion toward higher terraces and low hill areas, the thickness distribution of Quaternary overburden, and the physical properties of rocks and soil vary greatly. All of these lead to an uneven pattern of settlement. |
| Natural reserve (NR) | LS is a natural and cultural heritage city with many scenic spots, nature reserves, national forests, and geological parks. All of these play an important role in regional economic development and ecological resources protection. |
| Landscape type (LST) | Land use and cover was reclassified into six types (woodland, grassland, farmland, water area, constructed land, and bare land). The basic farmland, green space (e.g., mountain forest parks), and nature reserves are unsuitable for expansion and so the resistance is infinity (ecological barrier). In addition, we allocate resistance values based on the location of grasslands and woodlands and their ecological values. |
| Density of the urban population (DUP) | The urban population is the driving force of urbanization. The higher the density of the urban population, the lower the resistance to urban expansion, and vice versa. |
| GDP | GDP reflects the development level of the regional economy, as well as regional input and output. It shows the vitality of various type of production and construction in the region. |
| Convenient transportation (CT) | Compared with other types of cities, the road networks in mountain cities are more conducive to urban expansion. Therefore, road network data are processed to analyze the convenience of transportation and urban expansion resistance. |
| Water supply-drainage conditions (WSDC) | There are dense river networks often affected by floods. Accordingly, WSDC was select as a resistance factor of UE. |

Step 2: Grading and Assignment of ESRF

The grid values and units of the 10 resistance factors are different. For example, the elevation ranges from 147 to 641 m, while the slope is measured in degrees (°). Moreover, LST and NR have no units. Therefore, it is meaningless to conduct overlay analysis. To solve this problem, we needed to reclassify and assign the resistance factor data (Table 2), and convert the data into a comparable and dimensionless grid. There are five types of landscapes in LST, and we reclassified the other resistance factors into five categories to avoid null values due to excessive or insufficient categories. Moreover, the principle of assignment is the simple data structure and convenient calculation, because this assignment is dimensionless and does not affect the calculation results.

**Table 2.** The grade and assignment of resistance factors, and calculation results of resistance factor weight by logic regression analysis and normalization.

| Resistance Factors | Sub-factors | Level | Assignment | Weight |
|---|---|---|---|---|
| Terrain | Elevation (m) | <374 | 1 | 0.11 |
| | | 374–400 | 3 | |
| | | 400–431 | 5 | |
| | | 431–478 | 7 | |
| | | >478 | 9 | |
| | Slope (°) | <6.5 | 1 | 0.07 |
| | | 6.5–11.5 | 3 | |
| | | 11.5–18.5 | 5 | |
| | | 18.5–27.5 | 7 | |
| | | >27.5 | 9 | |
| Engineering Geology | GH (Freq.) | <1 | 1 | 0.11 |
| | | 1–2 | 3 | |
| | | 3–4 | 5 | |
| | | 5–6 | 7 | |
| | | >7 | 9 | |
| | SBC (t/ m$^2$) | >70 | 1 | 0.09 |
| | | 55–70 | 3 | |
| | | 40–55 | 5 | |
| | | 25–40 | 7 | |
| | | <25 | 9 | |
| Urban Ecology | LST | Built-up area | 1 | 0.13 |
| | | Grassland | 3 | |
| | | Farmland | 5 | |
| | | Woodland | 7 | |
| | | Water bodies | 9 | |
| | NR | Ecological barrier | +∞ | 0.06 |
| Socioeconomic | DUP (sq. km) | >3000 | 1 | 0.11 |
| | | 200–3000 | 3 | |
| | | 1000–2000 | 5 | |
| | | 500–1000 | 7 | |
| | | <500 | 9 | |
| | CT (km) | <1 | 1 | 0.1 |
| | | 1–2 | 3 | |
| | | 2–3 | 5 | |
| | | 3–4 | 7 | |
| | | >4 | 9 | |
| | WSDC (km) | <1 | 1 | 0.1 |
| | | 1–5 | 3 | |
| | | 5–0 | 5 | |
| | | 10–15 | 7 | |
| | | >15 | 9 | |
| | GDP (¥) | >100,000 | 1 | 0.12 |
| | | 50,000–100,000 | 3 | |
| | | 30,000–50,000 | 5 | |
| | | 10,000–30,000 | 7 | |
| | | <10,000 | 9 | |

Step 3: Logic Regression Analysis

With the aim of exploring the authentic relationship between built-up areas and ESRF, and to obtain the weight of RFs, we established the logic regression relationship between them, and then

calculated the logic regression coefficient. The regression coefficient indicates the degree of influence of each RF on the development of the city in the past. In the future, the process of urban expansion will be affected by the same factors with a similar resistance. For example, crossing mountains and rivers with urban construction will increase costs. The fitting effect and significance of the logic regression model is different in various scales, and therefore, the RFs DRG data (30 m) after grading and assignment were resampled into five resolutions: 30 m (1 pixel), 90 m (3 pixels), 150 m (5 pixels), 210 m (7 pixels), 270 m (9 pixels). Through training and experiment, we found that the logic regression analysis with 90 m resolution passed the relative operating characteristics (ROC) curves test. Therefore, the DRG data of this scale was taken as the most suitable scale for the evaluation of urban expansion. The specific formulation is as follows:

$$\text{Logit} \ (p_i) \ = \ \beta_0 + \beta_1 x_1 + \beta_2 x_2 + \beta_3 x_3 + \cdots \beta_m x_m, \tag{2}$$

where $p_i$ is the probability of urban expansion without considering the distance to the source; $\beta_0$ is the constant of the regression model; $\beta_1 \cdots \beta_m$ are the regression coefficients; and $x_m$ (m = 10) is the resistance factor.

To calculate the LRS, we explored the relationship between the expansion source and RFs and obtained the regression coefficient based on logic regression analysis. The regression coefficient after normalization (Formula 3) is particularly vital because it is the weight of the overlay analysis, which we use to calculate LRS in the following process. It should be noted that this weight is different from that obtained through the AHP method, which is based on the experiences or judgments provided by one or more experts. The normalization formula is as follows:

$$w_i = \frac{\beta_i}{\sum_{i=1}^n \beta_i}, \tag{3}$$

where $w$ is the weight of the overlay analysis and the $\beta$ is the regression coefficient. $n$ ($n = 10$) represents the number of RFs.

Step 4: Calculation of the LRS

Through the processing and calculation of the above three steps, we obtained the RFs of the best scale and the weight of each resistance factor. Overlay analysis was performed to calculate the basic surface (BS). However, the problem of the homogenization of the sources was still not solved. To do this, the expansion source was divided into urban core areas, urban surrounding areas and satellite towns according to the intensity of urban land development (the ratio of construction land area to total land area), GDP, and DUP. These three divisions were named Level 1, Level 2, and Level 3, respectively. The goal of this step was to simplify the data structure and facilitate the calculation. Moreover, since the resistance value of built-up areas was assigned to 1 in Table 2, and urban core areas have a stronger expansion capacity than satellite towns, level 1, level 2, and leve1 3 were assigned values of 0.8, 0.9, and 1, respectively. Finally, we used the formula LRS = BS × level to calculate the LRS.

(3) Calculation and Analysis of the LMCR Surface

The suitability evaluation, simulation of expansion direction, and connectivity analysis are all based on the LMCR surface. We used the source data and LRS to calculate the LMCR using the cost distance method. We used a digital grid (Figure 7) to analyze the three questions posed in the introduction section.

(1) To evaluate the suitability of urban expansion, we used the LMCR surface to conduct suitability zoning (the reclassification method); thus, we achieved the suitability distribution of urban expansion (Figure 7B).
(2) We extracted the resistance curve of the LMCR surface from the urban geographic center (UGC) to satellite towns to evaluate the resistance variation in different directions (Figure 7C).

(3)　　The expansion path and direction were calculated from the LMCR surface using cost path analysis (Figure 7D). R represents the value of the LMCR from landscape unit $d$ to source unit $a$. $R_{abd}$, $R_{ad}$, and $R_{acd}$ are defined as the expansion path.

$$R_{abd} = \frac{2+2}{2} + \frac{2+5}{2} + \frac{5+6}{2} + \frac{6+4}{2} + \frac{4+5}{2} + \frac{5+6}{2} + \frac{6+8}{2} + \frac{8+9}{2} + \frac{9+7}{2} + \frac{7+8}{2} + \frac{8+9}{2} + \frac{9+7}{2} = 70.5$$

$$R_{ad} = \frac{(2+3) \times \sqrt{2}}{2} + \frac{(3+2) \times \sqrt{2}}{2} + \frac{(2+6) \times \sqrt{2}}{2} + \frac{(6+9) \times \sqrt{2}}{2} + \frac{(9+7) \times \sqrt{2}}{2} = 24.5\sqrt{2} \qquad (4)$$

$$R_{acd} = \frac{2+2}{2} + \frac{2+1}{2} + \frac{1+1}{2} + \frac{1+1}{2} + \frac{1+2}{2} + \frac{2+1}{2} + \frac{1+2}{2} + \frac{2+1}{2} + \frac{1+8}{2} + \frac{8+7}{2} = 22.5$$

As we see from the above calculations, the minimum resistance value is $R_{acd}$, and thus the most suitable expansion path for the above three paths is $R_{acd}$. Moreover, we combined the urban planning of LS in 2020 and 2030; during these periods, the urban scale is controlled at 100 km$^2$ and 140 km$^2$ to balance the constructed land and urban green space and realize a sustainable urban development. In view of this, four scenarios were designed, namely, 80 km$^2$, 100 km$^2$, 140 km$^2$ and 160 km$^2$, to analyze the connectivity of urban–rural areas and the protection of urban green space. The expansion source and ecological barriers were used as input data for the buffer analysis, and the buffer scale was set to the four scenarios listed above.

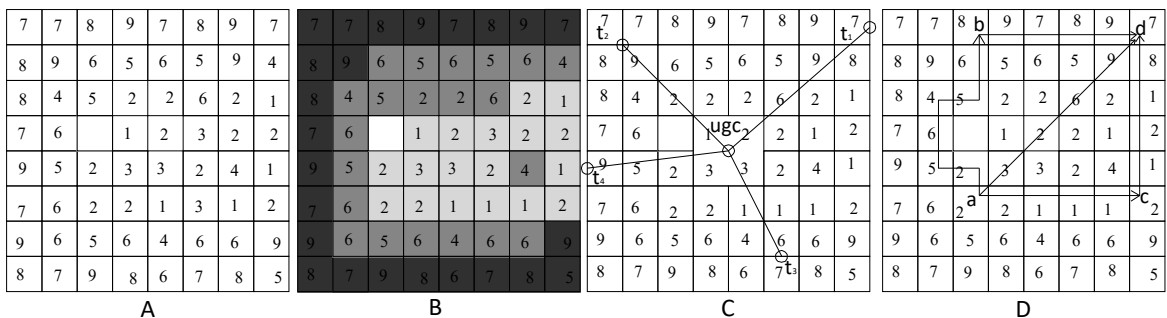

**Figure 7.** The digital grid consists of the basic cell representing the LMCR value, and the cell with no data denotes the ecological barrier. The four digital grids (**A**, **B**, **C**, and **D**) represent the LMCR surface, suitability zone, the resistance variation in different directions and the calculation of expansion paths, respectively. Moreover, the urban geographic center (UGC) denotes the urban geographic center and $t$ indicates the satellite town.

## 3. Results

### 3.1. Resistance Analysis of UE

Some important satellite towns (e.g., Suji, Mianzhu, Juzi and Mouzi) and villages surround the central district of LS and constitute its urban–rural gradient structure [31–33]; the connectivity of these towns has increased gradually with the increases in economic exchanges. However, we found that the expansion resistance was significantly different from the central district to each satellite town based on the LMCR surface. This indicated the differences in conflicts and competitions between the constructed land and other landscape units in the various expansion directions. Particularly, the resistance increased dramatically when UE crossed the rivers or hills in the expansion direction. A resistance trend analysis was performed from the UGC to satellite towns, and the results are shown in Figure 8. Although the satellite towns had the same level of development (e.g., GDP and DUP), the variation trends of resistance were different. From UGC to Suji, the expansion process bypassed the Green Heart Park and crossed two rivers (important resistance factors) and led to an increase in the resistance (Figure 8a). The distance from Mianzhu to UGC was 20 km, and the average resistance value in the Mianzhu direction was lower than the resistance value of the Suji direction; this is because no rivers or hills were crossed during the expansion process. Moreover, the central district and Mianzhu are connected

by 305 provincial roads and many villages distributed along the roads. This also promoted urban expansion. In the southward expansion to Juzi (LS high-tech zone), apart from bypassing Green Heart Park and crossing the Dadu River, UE overcame less resistance. In addition, this area had a flat terrain and developed industry. Figure 8c shows that the resistance increased only when bypassing Green Heart Park and crossing Dadu River. Northward to Mouzi, the dense urban population and flat terrain provided indispensable conditions for urban expansion, accordingly, the resistance curve was generally smoother than in the other regions (Figure 8d). Finally, Mouzi had convenient transportation, such as the beltway and the highway connecting it to Guanmiao.

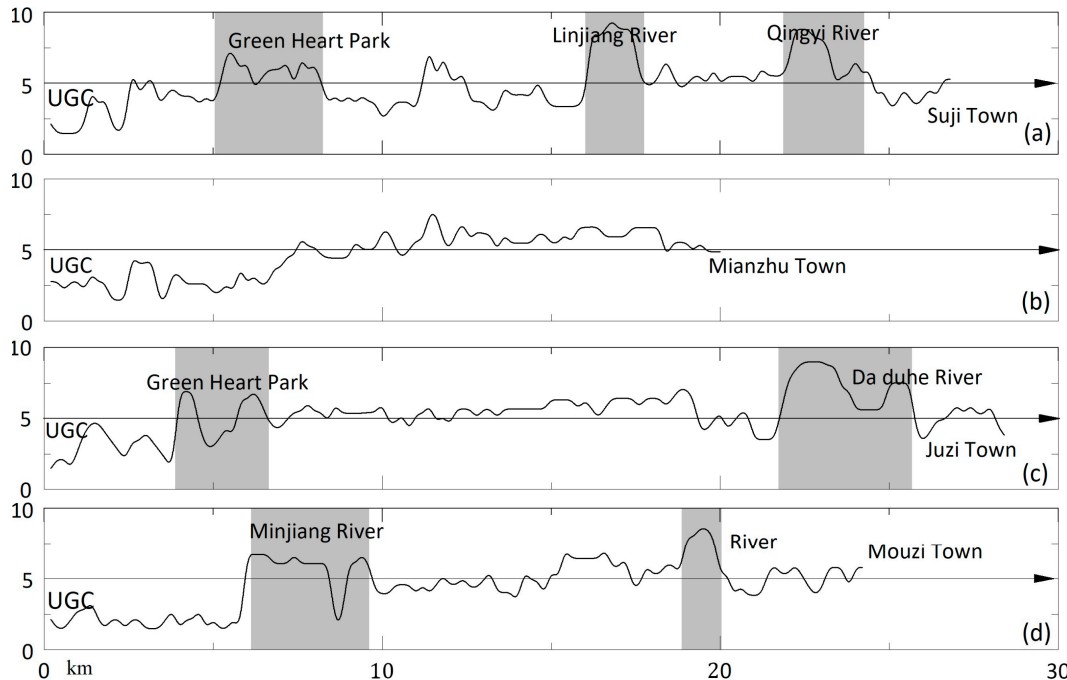

**Figure 8.** Resistance curves of urban expansion. (**a**), (**b**), (**c**), and (**d**) are the resistance curves from UGC to Suji, Mianzhu, Juzi, and Mouzi, respectively.

*3.2. Suitability Evaluation of UE*

To evaluate the suitability of urban expansion, LMCR surface (Figure 9) zoning was performed and the whole area of LS was divided into three types according the value of the LMCR surface. Specifically, the three zones were the expansion source, suitable expansion zone, and unsuitable expansion zone, respectively (Table 3). The proportion of expansion source area was 9.92% and the value of the LMCR surface was 0, including built-up areas and urban planning areas. Examples of expansion source areas are Boshui (BS) street, Buddha street, Zhanggongqiao (ZGQ) street, Shanghe (SH) street, Boyang (BY) street, Tianjian (TJ) street, and Xiaoba (XB) street. The proportion of the expansion zone that was suitable was 50.8% and the range of LMCR was 0 to 9733. The suitable expansion zone was divided into two levels. The area of level 1 accounted for 23.5% of suitable expansion zone and it was mainly distributed in Mianzhu, Juzi, Suji, and Mouzi. There was convenient transportation, regular terrain, and a high subgrade bearing capacity in this area. Moreover, the provincial highway near Mouzi was a channel for the supply of goods and raw materials for the central district; meanwhile, Mianzhu town is located in a key position along the LS-Chengdu expressway, and therefore, the convenient traffic around Mianzhu promoted urban expansion. Some villages were distributed in the southward expansion into Juzi and then into the Wutongqiao district. The level 2 zone accounted for 27.3% of the suitable expansion area and it was mainly distributed in the north of Mianzhu and Mouzi, Guanmiao, and the Shawan district. Although this area was affected by floods to a small extent in specific months, it was suitable for expansion as indicated by some engineering reinforcement measures. In addition,

the terrain is relatively flat and close to the highway. The area of unsuitable urban expansion zone (core zone and buffer zones) was 331.7 km$^2$. The core and buffer zones accounted for 10.2% and 29.1% of the total unsuitable expansion area, respectively. The LMCR value of the core zone was from 30,633 to 47,000 and it was distributed in LS Buddha and the surrounding region, mountain forest parks, urban ecological green land, Minjiang water-source reserves, basic farmland protection areas, and protected mountain ecological areas. The buffer zones were the transitional areas from the core zone to a non-core zone, e.g., the buffer zone of a nature reserve, scenic spot, mountain forest park, water source protection area, and ecologically sensitive area. The LMCR value of the buffer zone was 9733 to 30,633 and the total area was 85.2 km$^2$.

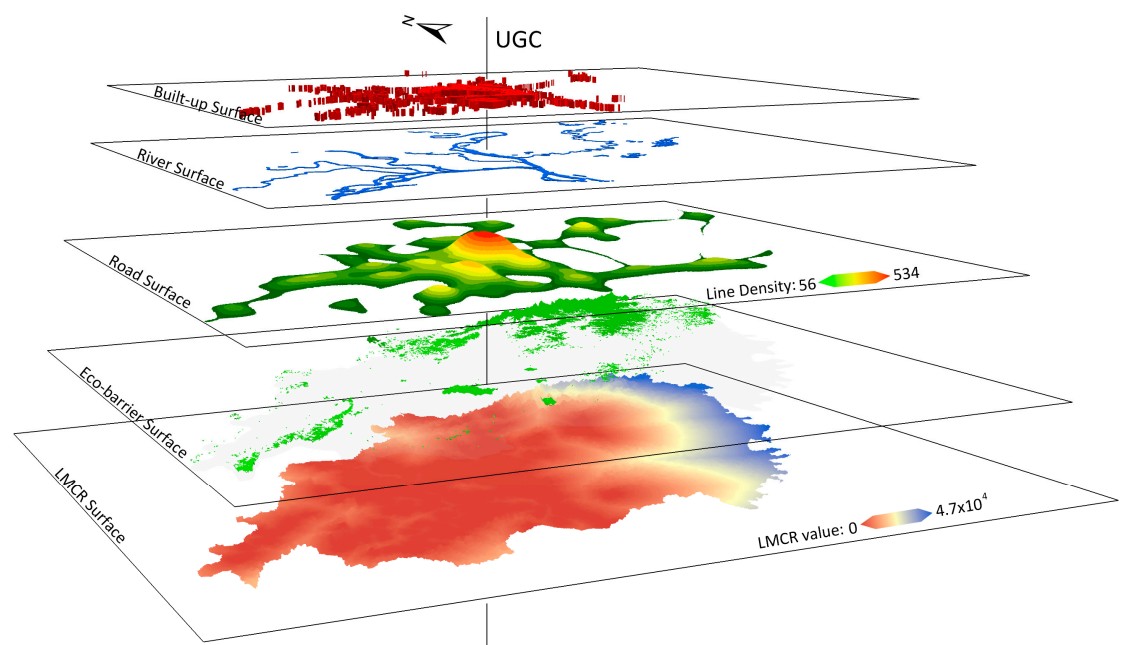

**Figure 9.** Overlay layers of the LMCR surface, eco-barrier, and RF surfaces.

**Table 3.** Suitability zoning and corresponding basic information.

| Zone | Sub-zone | Percent | LMCR | Distribution |
|---|---|---|---|---|
| Source | Source | 9.9% | 0 | Boshui (BS) street, Buddha street, Zhangogngqiao (ZGQ) street, Shanghe (SH) street, Boyang (BY) street, Tianjian (TJ) street, and Xiaoba (XB) street. |
| Suitable expansion | Level 1 | 23.5% | 0–4570 | Mianzhu, Juzi, Suji, and Mouzi. |
| | Level 2 | 27.3% | 4570–9733 | The north part of Mianzhu, including Mouzi, Guanmiao town and Shawan district. |
| Unsuitable expansion | Buffer | 29.1% | 9733–30633 | Buffer zones of nature reserves and scenic spots. |
| | Core | 10.2% | 30633–4700 | LS Buddha, mountain forest park, Minjiang water-source reserves, basic farmland, and protected mountain ecological areas. |

### 3.3. Simulation of Direction and Scale of UE

Green Heart Park, also known as Eco-Green Heart, is a beautiful tourist spot in LS city; thus, this area was strictly protected during UE. Eco-green heart was regarded as an eco-barrier and was not crossed during UE. Other eco-barriers included water-source reserves, basic farmland, green space (e.g., eco-park, greenbelt), and ecologically sensitive areas. In Section 3.1, we analyzed the resistance of UE from the central district to the satellite towns; the resistance value was low in the south, west, and north directions. Conversely, the urban expansion resistance was high in the east direction and was limited by natural and social conditions. We calculated the expansion path from UGC to satellite towns in the south, west, and north directions. The expansion path was extracted from the LMCR surface using the cost path method. The line width and length represented the expansion power and distance, respectively (Figure 10). As shown in Figure 10, there were four directions from UGC to satellite towns: the Mouzi direction (northeast), Mianzhu direction (northwest), Suji direction (west), and Juzi direction (south). The expansion paths in each direction were as follows. To the northeast, urban expansion crossed the Minjiang River to the northern part of Mouzi, and to Guanmiao; finally, the expansion connected to the town of Tuzhu. The northwest direction was more suitable for urban expansion due to the low resistance and fewer eco-barriers. In this direction, along the Zhugong river, Mianzhu was connected to the central district. In view of the expansion capacity and scope (see Figure 10), this path was indispensable for the urban expansion of LS; the expansion level was classified as level 1. A high-speed railway station and the new Qingjiang (QJ) district, which is under construction, are in this area, and the convenient transportation, dense urban population, and fewer eco-barriers promoted its urban expansion. Westward expansion bypassed Green Heart Park to Suji, and extended to Emeishan, a county-level city of LS. There was another path expanding to the town of Yangwan along the Linjiang River, but this expansion path had a low expansion capacity. To the south, urban expansion crossed the Dadu River to Juzi, and connecting to the Wutongqiao district.

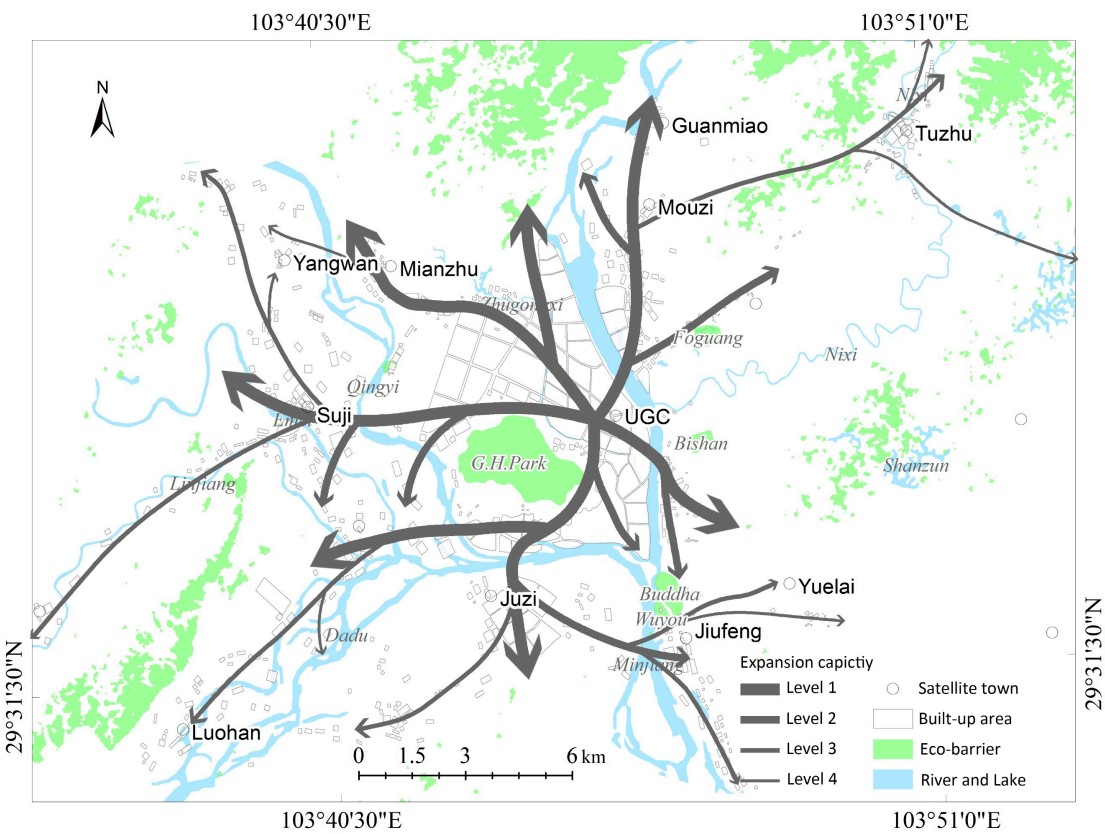

**Figure 10.** Directions and paths of urban expansion in LS.

According to the urban planning of LS, the scale of constructed land in the central district will be controlled within 100 km$^2$ and 140 km$^2$ in 2020 and 2030, respectively. With the expansion of the urban scale, the connectivity between the central district and the surrounding area will be more convenient. However, the green space, especially garden plots and grassland will be compressed. Considering this, we analyzed the connectivity of the urban–rural areas and the protection of green space in different urban scale scenarios. Therefore, four scenarios were designed to refer to the scale of urban planning of LS: 80 km$^2$ (scenario 1), 100 km$^2$ (scenario 2), 140 km$^2$ (scenario 3), and 160 km$^2$ (scenario 4). Scale simulations of UE in the different scenarios were performed, and the results are shown in Figure 11. All of the satellite towns, except for Tuzhu and Lingyun, interconnected in scenario 2. This indicates a high connectivity between the central district and satellite towns under the 120 km$^2$ urban scale. In addition, the size of the green space is suitable. In scenario 3, the important satellite towns (e.g., Mianzhu, Suji, Juzi and Mouzi), are incorporated into the central district, and the urban scale reaches 140 km$^2$. As a result of the urban expansion, some green space and basic farmland are occupied and converted into constructed land. With the urban scale continuing to expand, in scenario 4, the urban scale reaches 160 km$^2$ and the urbanization rate is 86.1%. However, most of the green space, basic farmland, and ecologically sensitive areas are occupied by constructed land. Such a development model is unsustainable and not recommended. Following comprehensive analyses of the above four scenarios, the most suitable scale is 132 km$^2$ for LS. In this scale, LS not only has high connectivity between urban and rural areas, but fewer green spaces, basic farmland areas, and nature reserves become occupied.

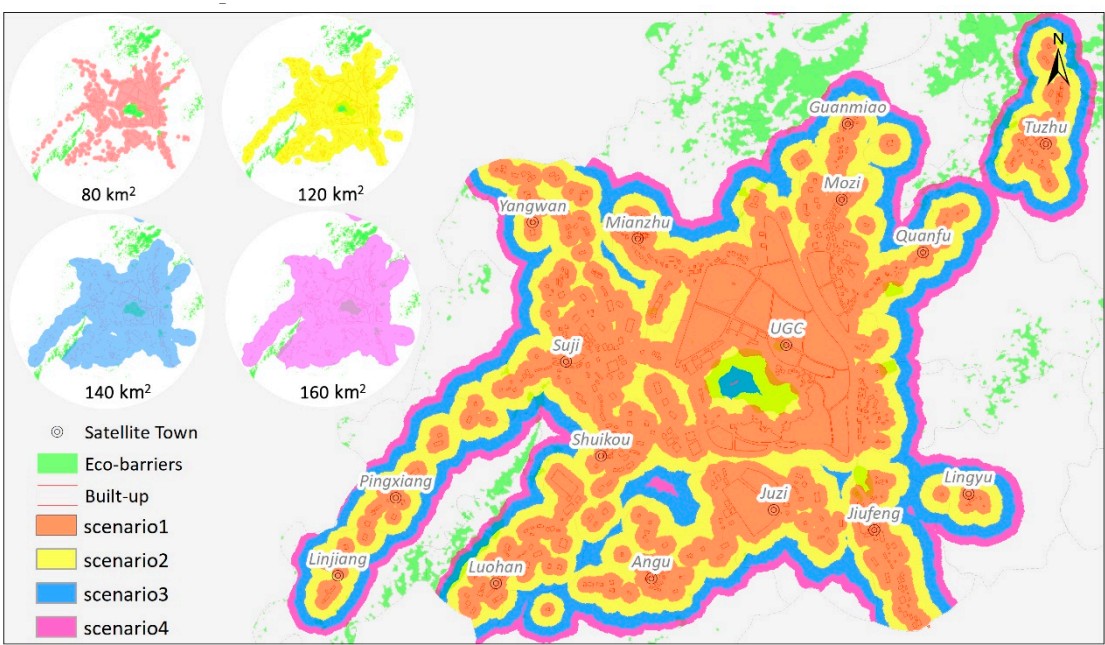

**Figure 11.** Connectivity and scales in the different scenarios.

## 4. Discussion

### 4.1. Validation of Results

We established the ESRF to evaluate the resistance of UE, and then used the LMCR model to simulate the direction and scale of UE. The results showed that the directions of Mianzhu, Guanmiao, Suji and Juzi were the most suitable for expansion, and they had lower resistance from other landscape units. In 2030, some important satellite towns will be incorporated into the central district with through urban expansion. We discussed the simulation results with experts of the LS Urban Planning Bureau (LUPB) for the sake of validating the results and the accuracy of the LMCR model. Fortunately, they

gave their support and we obtained the 2017 version of the urban planning map (Figure 12) of LS, which was approved by the government and well praised by the public. The urban development plan from 2018 to 2030 was elaborated in this version, including the urban development mode, direction, and scale of UE and ecological protection measures. The ecological red line (ecological barriers) was clearly elaborated in the ecological protection method, including the protection of basic farmland, water sources, and green space, which were the ecological barriers in this case study. From the planning map, it was obvious that the directions of UE were northward, westward, and southward. Moreover, many garden plots and parks were preserved along the river. This was consistent with the simulation results of UE in this case study (Figure 10). In scenario 3 (Figure 11), the simulation results were compared with the urban planning map. It can be seen that the scale of UE was consistent with the planning map. For validation, we compared the area of constructed land in scenario 3 and the planning area of 2030 by calculating the ratio of area; the ratio of calculating was 0.9817. This ratio showed that the error between the scale of simulation and planning was 1.83%. Moreover, Suji, Mianzhu, Juzi, and Mouzi were incorporated into the central district in 2030.

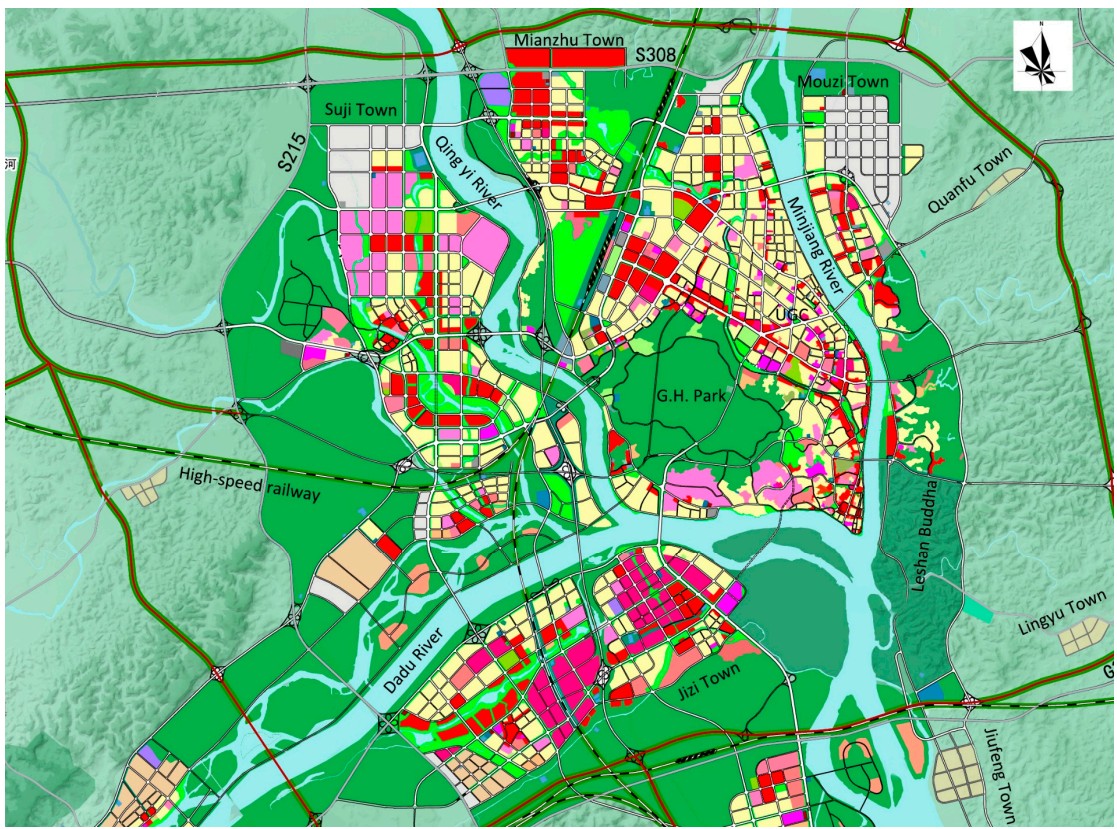

**Figure 12.** Urban planning map of LS from 2018 to 2030 (2017 version, LS government).

In the urban expansion of LS, according to the characteristics of mountain cities and referring to the UEPS, the scientific ESRF of UE was established. Moreover, the resistance surface was improved and the problem of homogenization of expansion sources was solved based on logic analysis and cost path analysis. The simulation results were in agreement with the urban planning of LS (2017 version). However, we need to point out that no model or method is perfect. UE is a complex process accompanied by conflicts between the social economy and the natural environment, and RFs are diverse and variable across different areas. With the improvement of construction technology, the adjustment of government policy, and the change of people's attitudes towards nature, there are many possibilities for urban development in the future. In this case study, there were 10 RFs in ESRF, and we explored the relationship between RF and UE through logic regression analysis. Occasionally, the

process of UE did not completely follow such a logical relationship, and there were some deviations. Specifically, the transportation conditions (CT) were not convenient enough in some areas, but the land prices were lower than in other areas. Thus, such areas will become suitable expansion zones for UE. In addition, the weights of the two factors (CT and land prices) were similar, but it is possible that only one of them will play a decisive role in the actual UE. This problem has been mentioned in various case studies [34,35]. The evaluation factors of previous studies [21,35–37] regarding urban expansion and the RFs of UE in this case study were of about the same number. Some new RFs could be added to ESRF, such as policy factors and decision-making preferences, but how to load them into the model remains to be studied. In future works, long-term series remote sensing images should be used to better explore the relationship between UE and RFs. There are some good references available [18,38–40] related to the use of remote sensing data for urban expansion. Compared with traditional models, such as CA, Clue-s, GRNN, SLEUTH, AHP, and OWA, LMCR is more efficient and scientific. However, coupling of models could be used to evaluate and simulate urban expansion and/or land use and cover changes (LUCC)in the future, allowing one model to verify the other, thus, improving the accuracy of the results.

*4.2. Strategies for UE*

The level 1 and level 2 areas accounted for 23% and 27.3% of the suitable expansion, respectively. We should give priority to expansion level 1 because large-scale urban construction and human activities will aggravate urban ecology and cause soil erosion in level 2. We propose that LS expand westward so as to bypass low hilly areas and areas with a high risk of geological hazards, eventually connecting with Emeishan. According to the urban development orientation (an ecotourism city) of LS, it is suggested that 132 $km^2$ is the most suitable urban size. Additionally, we should protect vegetation, basic farmland, etc., and avoid polluting the three rivers (the Minjiang River, Qingyijiang River, and Dadu River) through urban expansion and human activities as it will have a negative impact on downstream areas. In addition, development of the unsuitable expansion zone should be strictly prohibited. Residents who live in these areas, especially in low hilly areas, protected water resource areas, and nature reserves, should be relocated. The nature reserve buffer zones, particularly around the Grand Buddha Scenic Area and Green Heart Park, have high intensity human activity and high ecological fragility, and because of this, the existing protection measures should be improved.

We should balance the construction land and green space and strictly control the occupancy of green space. Taking the three riverbanks (the Minjiang River, Qingyijiang River, and Dadu River) banks and Green Heart Park as the core, we should better plan and protect ecological spaces. In addition, urban renewal should be focused and considered, rather than following the path of blindly building new districts. Moreover, it is necessary to develop the satellite towns around the central district and promote connectivity between the satellite towns and the central district, thus constructing a rational urban-rural gradient structure.

## 5. Conclusions

Using the LMCR model, we combined the high spatial resolution of ZY03 satellite images with statistical and monitoring data to evaluate the urban expansion in a typical mountain city. The focus of the research was the suitability evaluation of construction land, and the direction and connectivity of UE in LS. The acquired findings on urban expansion in this mountain city can be summarized as follows. We explored the relationship between the RFs and UE by logic regression analysis and obtained the regression coefficients (weights). The weights indicated that landscape type, economic development, and geological hazards had an important impact on urban expansion for this mountain city. Through suitability zoning, we found that the proportion of suitable urban expansion areas was only 23.5%, and the unsuitable expansion areas was 39.3%. The area of suitable expansion (level 1) was less than in other types of cities (this is a common problem for the development of mountain cities). Therefore, we could efficiently develop the level 2 zone through environmental protection

and engineering reinforcement measures. The suitable directions of UE were southward, westward, and northward. The expansion resistance towards Mianzhu was the lowest, and thus it was the most suitable expansion direction. In addition, we found that setting up the ecological barriers makes the expansion direction more reasonable in a mountain city. The suitable scale of urban expansion of LS is 132 km², and when the urban scale expands to 140 km², Mianzhu, Guanmiao, Suji and Juzi become e incorporated into the central district.

Through this case study, we found that the LMCR model is a feasible method of analysis and has a high precision in terms of simulating urban expansion in mountain cities. Thus, we can approve the use of the LMCR model in other mountain city scenarios as a general method at various scale. Moreover, the use of this model should be based on a scientific evaluation system (ESRF), and the evaluation system must be comprehensive and conform to the special conditions of mountain cities.

**Author Contributions:** Conceptualization, H.W. and P.P.; Methodology, H.W. and T.Z.; Software, X.K.; Validation, H.W. and X.K.; Formal Analysis, H.W. and G.Y.; Investigation, X.K.; Resources, H.W.; Data Curation, H.W. and X.K.; Writing-Original Draft Preparation, H.W.; Writing-Review & Editing, H.W. and X.K.; Visualization, H.W. and T.Z.; Supervision, H.W. and P.P.; Project Administration, H.W. and T.Z.; Funding Acquisition, H.W. and P.P.

**Funding:** This research was funded by the National Key R&D Program of China (NO. 2017YFC0505001), National Natural Science Foundation of China (NO. 41501060), and the University Fund Project of ETC (NO. C122017007).

**Acknowledgments:** We acknowledge CCRSDA, WESTDC and USGS for the freely downloadable ZY03 satellite remote sensing image, LUCC map products and SRTM DEM, respectively. We would like to thank the anonymous reviewers for constructive and detailed comments.

**Conflicts of Interest:** The authors declare no conflict of interest.

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
