# Peer review of "Evaluating the Suitability of Urban Expansion Based on the Logic Minimum Cumulative Resistance Model: A Case Study from Leshan, China"

_ijgi, doi:10.3390/ijgi8070291_

Round 1
Reviewer 1 Report
It is a meaningful research work, and the paper has been greatly improved after several rounds of revisions. Just one suggestion.
1. The authors should add some general evaluation of the method in the conclusion part, to clarify the values of the research work. Not just the merits, but also the shortages.
Author Response
Thank you for your constructive comments in this minor revision. We analyzed and discussed again, and responded to your concerns. Moreover, the manuscript was proofed again by a native speaker of English.
Q1. The authors should add some general evaluation of the method in the conclusion part, to clarify the values of the research work. Not just the merits, but also the shortages.
Answer: It’s a good question, we agree with you. The conclusion section was revised and added the general evaluation of the LMCR model, including the merits and shortages. Please see the conclusion part of the main text.
Reviewer 2 Report
The authors have made a significant improvement to the document.
1) I suggest to change the title of step 3 to "Linear regression".
2) Figure 9 has a poor resolution,
Author Response
Thank you for your constructive comments in this minor revision. We discussed again, and responded to your concerns. Moreover, the manuscript was proofed again by a native speaker of English.

Reviewer 3 Report
Dear authors,
I read this new revised version of your paper and really appreciated the efforts you made. As I requested, you significantly improved your paper, especially in describing the adopted methods. Moreover, also the style of this manuscript is improved, therefore I definitely agree to publish it in its present form.
Author Response
Thank you for your recommending the manuscript to publish in IJGI. As I promised, the manuscript was proofed again by a native speaker of English. Professor, thank you very much.
This manuscript is a resubmission of an earlier submission. The following is a list of the peer review reports and author responses from that submission.
Round 1
Reviewer 1 Report
The authors proposed a LMCR model to simulate the urban expansion in a mountain city, Leshan, in China. Generally, the paper had analyzed the expansion direction of the city under constrained conditions and discussed connectivity between the central district and satellite town, and I think it was the core of innovation of paper. However, the key point had not be well presented.
1) The title should be rewritten. Is it a methodology paper or case study paper?
2) The abstract of the paper was too specific. I think the authors should give more general conclusions in this part. How was the study referable to other mountain cities?
3) The literature review was very weak. I think the authors should give a more systematic review of the researches concerning city expansion, land use simulation (e.g. cellular automation), and constrained development of mountain cities.
4) Most of the use of reference in this paper did not conform to the standards. For example, the citation of reference [1], a source of statistics, should be directed to a set of digits, rather than a general description of the status.
5) The citation of Figures should be more specific. For example, the Fig.1 and Fig. 2 should be cited separately, at end of the most related sentences respectively.
6) For the materials part, Section 2.2, we suggest the authors to give a more detailed introduction of the data preprocessing procedure, and at least one figure is needed here.
7) In line 127, the authors said “It’s very different for the resistance factors selected in the different city”. So, they should elaborate why the 10 resistance factors were selected in this paper, not just list them out and say it was in accordance to some materials.
8) All methods have their hypothesis, especially for simulation algorithms. In Section 2.3, the authors should clearly describe the hypothesis of the paper and why the method proposed was effective here.
9) In Section 2.3.2, function (1) was a linear regression model, not a Logic regression model. The authors should clearly describe the whole function, not just the parameter part.
10) In Section 2.3.3, the construction of resistance surface is an importance part of the whole method introduced in this paper. The authors should elaborate this section, and at least give a figure to illustrate the relationship between the resistance surface and the 10 resistance factors.
11) In Section 2.3.4, the LMCR model is the core of the paper. However, it is so simple. It is hard to understand the function. Meanwhile, the formation of function (2) did not conform to the standards. What “fmin” means? There is not such style of writing.
12) It is suggested to add a table in Section 3.2 for illustration of suitability evaluation of UE.
13) The authors should give a figure to depict the LMCR surface before the description of direction and path of urban expansion in Section 3.3.
14) In Section 3.3, it is suggested the authors to clearly listed the 4 scenarios of simulation, and of course give full discussion on the differences and variations of the 4 scenarios.
15) The discussion part Section 4 should be more specific on the analysis results. How was the accuracy of simulation? To what degree, the results was effective?
16) The conclusion part should be Section 5, not Section 4.
Author Response
Dear Reviewer
I am very appreciating for your comments and suggestions. My team and I spent 11 days carefully analyzing and revising the structure of the paper (method, result, discussion and conclusion)and especially paid much attention to your comments and suggestions. I would like to upload it to ISPRS International Journal of Geo-Information. Title of manuscript has been changed to “Evaluating the suitability of urban expansion based on LMCR model: A case study from Leshan, China” to make it clearer and sooth.
1. The title should be rewritten. Is it a methodology paper or case study paper?
Answer: Evaluating the suitability of urban expansion based on LMCR model: A case study in the LS city, China
2. The abstract of the paper was too specific. I think the authors should give more general conclusions in this part. How was the study referable to other mountain cities?
Answer: the abstract has been revised in the new manuscript.
Urbanization is accompanied by conflicts and intense transformation among various landscape that bring a series of social, economic, and ecological impacts. Evaluating the suitability of urban expansion (UE) and the scale of UE is vital to solving the urban environmental issue and realizing urban sustainable development. Taking Leshan (LS) city of China, a typical mountain city with rapid growth in upper Yangtze river, as the case study, the logic minimum cumulative resistance (LMCR) model was applied to evaluate the suitability of UE and simulate the direction and scale of UE. Results revealed that: The evaluation system of resistance factor (ESRF) should follow the principle of natural and social harmony in the mountain city, the logic resistance surface (LRS) is a scientific integration of multiple resistance factors based on the ESRF and logic regression analysis. LRS objectively reflects the contribution of each resistance factor to urban expansion, and landscape, geologic hazard and GDP have a great impact on the urban expansion of LS city. The expansion space of mountainous city is limited, the area of suitable expansion is only 23.5% and the unsuitable expansion area account for 39.3%. In addition, setting up the ecological barriers is an effective way to control the unreasonable urban expansion in the mountain city. There is an obvious scale (grid size) effect in evaluation of urban expansion in the mountain city, the suitable evaluation scale for urban expansion was 90 m x 90 m in LS city. On this scale, taking the central district as the center, the urban expansion process will extend to Mianzhu, Suji, Juzi and Mouzi town. Urban expansion should be controlled at a reasonable size, especially in mountain city, the most suitable urban size of LS city is 132km2 and regional towns had a high connectivity and few ecological lands occupied.
3. The literature review was very weak. I think the authors should give a more systematic review of the researches concerning city expansion, land use simulation (e.g. cellular automation), and constrained development of mountain cities.
Answer: The section has been revised.
4. Most of the use of reference in this paper did not conform to the standards. For example, the citation of reference [1], a source of statistics, should be directed to a set of digits, rather than a general description of the status.
Answer: sorry, I don’t understand what you said. Is the citation format of references incorrect? Doi?
5. The citation of Figures should be more specific. For example, the Fig.1 and Fig. 2 should be cited separately, at end of the most related sentences respectively.
Answer: Fig.1, Fig. 2 and their statements have been revised in the manuscript.
6. For the materials part, Section 2.2, we suggest the authors to give a more detailed introduction of the data preprocessing procedure, and at least one figure is needed here.
Answer: Data preprocessing (see fig. 5) was as follows: (1)Using the resources satellite No.3 image to update the land use and cover dataset (2011). And then we merged the subtype into a type, such as the sparse woods were merged into the wood, from this, we achieved five types of land use and cover of LS city. The specific type is woodland, grassland, farmland, water body and constructed land. (2) We extracted reservoirs, rivers, roads, built-up areas, pits, and greenbelts from remote sensing image (ZY03), the discrete data were converted to the continuous digital raster graphic (DRG) by buffer analyst, such as roads and rivers were transformed to the distance to roads and rivers, namely, the convenient transportation(CT) and the water supply-drainage conditions (WSDC). (3) Statistical and monitoring data, including the density of urban population (DUP), geological hazards (GH), gross domestic product (GDP) and subgrade bearing capacity (SBC) were interpolated into DRG by Kriging interpolation[25] method, and it’s a geostatistical analysis method, which is suitable for the interpolation of discrete point data. In view of the size of the study area and the spatial resolution of the raw data (DEM), the output raster resolution was set to 30m in buffer analysis and Kriging interpolation. In addition, all DRG were projected into UTM WGS 1984 reference system and clipped with urban bounder. The data preprocessing flow is shown in Figure 5.
7. In line 127, the authors said “It’s very different for the resistance factors selected in the different city”. So, they should elaborate why the 10 resistance factors were selected in this paper, not just list them out and say it was in accordance to some materials.
Answer: We comprehensively investigate and analyzed the natural and social environment of LS city. In details, LS is located in hilly area, with loose soil, dense river-net and rich precipitation, the conditions are prone to flood and geological disasters. Accordingly, geological hazards (GH), subgrade bearing capacity (SBC), water supply-drainage conditions (WSDC) are the important resistance factors affecting urban expansion; In the past 15 years, we witnessed LS city rapid economic development and urban population increase, which has accelerated urbanization (see figure 1). Therefore, from the perspective of social economy, the convenient transportation(CT), density of urban population (DUP), gross domestic product (GDP) are also important factors affecting LS’s development. Moreover, we also referred to the urban evaluation and planning standards (UEPS) [25] enacted by the Ministry of Housing and Urban-Rural Development of China (MHURD). Finally, we selected four categories including ten resistance factors to establish the ESRF. Specifically, there are elevation, slope, GH, SBC, WSDC, CT, DUP, GDP, natural reserve (NR) and landscape type (LST). The 10 resistance factors were divided into two types that one could be overcame (e.g., grassland) and the other one was not (e.g., ecological barriers, such as, natural reserve, water source area, basic farmland protected area, etc.). In details, the principle of resistance factor selection and its ecological analysis are shown in the table 1.
8. All methods have their hypothesis, especially for simulation algorithms. In Section 2.3, the authors should clearly describe the hypothesis of the paper and why the method proposed was effective here.
Answer: According to the principle of LMCR model to implement the model, we need to make four assumptions.
(1) Urban area is divided into two types by its use: constructed land, which is the land that is most suitable for urban expansion, and ecological land (including green space, rivers, lakes, basic farmlands, etc.), which is the land that is suitable for protection.
(2) Accelerating economic development and urban construction, the constructed land requires expansion. Conversely, to improve the quality of urban environment, ecological land requires expansion. The challenge of interaction in these two processes is to realize the dynamic balance between ecological land and constructed land.
(3) The unreasonable urban expansion can be corrected by setting up ecological barriers (e.g., green space, rivers, basic farmland, etc.) which in turn will restrict urban expansion. More over the ecological barriers effectively protect urban environment. That is to say, ecological land could be restrictive or driving factors in different processes.
(4) The urban expansion source is heterogeneous, additionally, whether an area is suitable for urban expansion is identified by the LMCR value under the urban expansion sources and various resistance
9. In Section 2.3.2, function (1) was a linear regression model, not a Logic regression model. The authors should clearly describe the whole function, not just the parameter part.
Answer: To explore the relationship between already existing built-up areas and 10 RFs, we established the logic regression relationship between them, and then we calculated the logic regression coefficient. Regression coefficient indicate the degree of influence of each RF on the development of the city in the past. In the future, the process of urban expansion will also be affected by the same resistance factors with similar conflicts, such as: crossing mountains and rivers in urban construction will increase costs. The fitting effect of the logic regression model is different in virous scales, therefore, the original DRG data (30m) were re-sampled into five resolutions, specifically, 30m, 90m, 150m, 210m, 270m. Through training and experiment, we found that the logic regression model with 90m resolution passed the relative operating characteristics (ROC) curves test. Therefore, the DRG data of this scale was taken as the most suitable scale for the simulation of urban expansion.
10. In Section 2.3.3, the construction of resistance surface is an importance part of the whole method introduced in this paper. The authors should elaborate this section, and at least give a figure to illustrate the relationship between the resistance surface and the 10 resistance factors
Answer: It’s a good idea. This part was revised in section 2.3.3.2.3 (Calculating the LRS)
11. In Section 2.3.4, the LMCR model is the core of the paper. However, it is so simple. It is hard to understand the function. Meanwhile, the formation of function (2) did not conform to the standards. What “fmin” means? There is not such style of writing.
Answer: The LMCR model is the core of this case study. Your comments are very important and constructive. I consulted the literature and discussed many times with my team, and revised the statement of the LMCR model strictly.
We divided a heterogeneous landscape into a “source” or “sink” landscape according to eco-process theory, of which the source landscape refers to those landscape that can promote the development of the process, and "sink" landscape refers to those landscape that can prevent or delay the development of the process. The qualities of "source" and "sink" landscape is relative, a "source" landscape in one process, it may be "sink" landscape in another process, accordingly, the analysis of "source" and "sink" landscape must be directed at a specific process. The source and sink landscape theory elaborated is to explore dynamic balance relationship between urban expansion and ecological protection, which in turn realize a suitable spatial pattern for urban expansion. The transformation between source and sink is a competitive process, which is performed through overcoming various resistance forces. And then all the resistance forces are integrated into a logic resistance surface. LMCR model essentially reflects the degree of resistance of urban expansion sources to sinks in the process of expansion. Of course, the degree of resistance varies with different sinks, which can be achieved by exploring the logical relationship between expansion sources and sinks. In addition, the cost distance in the model is not the actual distance, but the reflection of the spatial relationship between the two landscape units. We achieved the spatial relationship by calculating the drag coefficient when the expansion source passes through other landscape units. We can determine the connectivity between the two units based on the value of the minimum cumulative resistance, when the urban expansion source passes through a specific landscape, and usually the minimum cumulative resistance path is the suitable path of urban expansion.
12. It is suggested to add a table in Section 3.2 for illustration of suitability evaluation of UE.
Answer: This section has been revised and add the table 3.
13. The authors should give a figure to depict the LMCR surface before the description of direction and path of urban expansion in Section 3.3.
Answer: It has been revised, as shown in figure 7 in the manuscript
14. In Section 3.3, it is suggested the authors to clearly listed the 4 scenarios of simulation, and of course give full discussion on the differences and variations of the 4 scenarios.
Answer: Your suggestion is very constructive, in that case, we could analyze the connectivity clearly between the central district and satellite town. Figure 9 has been modified by recalculating and drawing, as shown in Figure 9.
15. The discussion part Section 4 should be more specific on the analysis results. How was the accuracy of simulation? To what degree, the results was effective?
Answer: The discussion section has been revised and we discussed the accuracy of simulation in this part.
16. The conclusion part should be Section 5, not Section 4.
Answer: Yes, section 4 has been revised to section5.

Reviewer 2 Report
The authors proposed Logic Minimum Cumulative Resistance model to combine high spatial resolution remote sensing images, environmental data and socio-economic data to evaluate the urban expansion in a mountain city.
Q1 : Some typos, line 100 : expansion; line 183 indicated , line 184 various etc …
Q2 : eq 1 seems to be a logistic regression. Since logic regression should be written using logic rules (eg. X1 or (X2 and X3)) etc …
Q3 : it is not clear why the authors are using 5 levels in line 156.
Q4: in table 4 what is the value of the intercept constant.
Q4: in figure 3, b1, b2, and b3 should be B1, B2, and B3
Q5: line 192, the sentence “ with an average value was bout 5” should be reformulated
Q6: line 230 reformulate the end of the sentence “with an area was 85.2km2”
Q7: the variation of the indices i and j in eq 2 are not clear …; what does k represents …
Q8: reformulate the sentence, from line 290
Q9: line 390 socioeconomic
Q10: I guess that it is possible to calculate the tests of significance of each Resistance Factor.
Q11: and to compare to other approaches and to calculate the concordance using for example the Kappa test.
Author Response
Dear Reviewer
I am very appreciating for your comments and suggestions. My team and I spent 11 days carefully analyzing and revising the structure of the paper (method, result, discussion and conclusion) and especially paid much attention to your comments and suggestions. I would like to re-submit it to ISPRS International Journal of Geo-Information. Title of manuscript has been changed to “Evaluating the suitability of urban expansion based on LMCR model: A case study from Leshan, China” to make it clearer and sooth.
Answers to your questions were as follows:
1. Q1: Some typos, line 100: expansion; line 183 indicated, line 184 various etc …
Answer: indicted—indicated, virous—various. Other misspelled words have also been revised.
Q2: eq 1 seems to be a logistic regression. Since logic regression should be written using logic rules (eg X1 or (X2 and X3)) etc.
Answer: it was revised, and specific formulate see response file (DOC)
Q3: it is not clear why the authors are using 5 levels in line 156.
Answer: The grid values and units of the 10 resistance factors are different, such as elevation data, which ranges from 147 to 641 in meters, while other resistance factors (SLOPE (°), GH (sq. km), SBC (t/m2), LST, NR, DUP (sq. km), CT (km), WSDC (km), GDP ()) are different, especially LST and NR have no the data units. Therefore, it is meaningless to calculate these 10 resistance factors by overlay analysis. To solve this problem, we need to reclassify and assign the resistance factor data (see table 2), and convert the data into comparable dimensionless values. There are five types of landscape in LST, accordingly we also reclassify other resistance factors into five categories to avoid null values due to excessive or insufficient categories
Q4: in table 4 what is the value of the intercept constant.
Answer: intercept constant=-8.6516
Q4: in figure 3, b1, b2, and b3 should be B1, B2, and B3
Answer: In order to better represent the logicality of the method, the fig. 3 has been deleted and redrawn.
Q5: line 192, the sentence “with an average value was about 5” should be reformulated
Answer: the sentence has been revised in the manuscript
Q6: line 230 reformulate the end of the sentence “with an area was 85.2km2”
Answer: the sentence has been revised in the manuscript
Q7: the variation of the indices i and j in eq 2 are not clear …; what does k represents …
Answer: see the DOC file (Attachme).
Where LMCR is the logic minimum cumulative resistance, f is a positive correlation function, which reflects the relationship between the minimum resistance and distance from one source (different grades) to another landscape in space and the characteristics of the resistance surface, denotes the minimum resistance overcame from a source j to a landscape unit i; is the spatial distance between landscape unit i and source unit j; and presents the resistance coefficient that exists in transition from landscape unit i to source unit j, is the level of source (e.g., the urban expansion source can be divided into the different levels according to its development degree).
Q8: reformulate the sentence, from line 290
Answer: the sentence has been revised in the manuscript
Q9: line 390 socioeconomic
Answer: the word has been revised in the manuscript
Q10: I guess that it is possible to calculate the tests of significance of each Resistance Factor.
Answer: The fitting effect of the logic regression model is different in virous scales, therefore, the original DRG data (30m) were re-sampled into five resolutions, specifically, 30m, 90m, 150m, 210m, 270m. Through training and experiment, we found that the logic regression model with 90m resolution passed the relative operating characteristics (ROC) curves test. Therefore, the DRG data of this scale was taken as the most suitable scale for the simulation of urban expansion.
Q11: and to compare to other approaches and to calculate the concordance using for example the Kappa test.
Answer: Thank you for giving me a good idea. We calculated the ratio of the scale of simulation and the area of planning and got the accuracy of the simulation, which is very similar to the Kappa coefficient. Specially, see the discussion section.

Reviewer 3 Report
I suggest to facilitate analysis of tavle and maps without the need to return to the text of the article. So for table 1 put a caption detailing the menaing of RF.
Please detail the MCR model and the improuvement due to LMCR.
Author Response
Dear Reviewer
I am very appreciating for your comments and suggestions. My team and I spent 11 days carefully analyzing and revising the structure of the paper (method, result, discussion and conclusion) and especially paid much attention to your comments and suggestions. I would like to re-submit it to ISPRS International Journal of Geo-Information. Title of manuscript has been changed to “Evaluating the suitability of urban expansion based on LMCR model: A case study from Leshan, China” to make it clearer and sooth.
Answers to your questions were as follows:
1. I suggest to facilitate analysis of tavle and maps without the need to return to the text of the article. So for table 1 put a caption detailing the meaning of RF.
Answer: To better read this section of the manuscript, we revised the caption of table1 (See table 1).
2. Please detail the MCR model and the improvement due to LMCR.
Answer: Through the analysis of previous research results using MCR model, we found that the model was not perfect and some limitation could be improved, especially the UE of mountain city. First, the weights allocation of resistance factors was still be conducted by expert scoring. Second, the homogeneous of expansion source was not solved. Viewed from this perspective, the MCR model would be inaccurate in evaluating urban expansion in mountain city, accordingly we improved this model and use it to evaluate the urban expansion of LS city. The specific improving process see the section 2.3.

Reviewer 4 Report
General comments
The paper “Using the LMCR model to evaluate and simulate the urban expansion in a mountain city, China” deals with the evaluation of effects of the Urban expansion in the LS city (China). As stated by the Authors, the logic minimum cumulative resistance (LMCR) model was applied to evaluate the suitability of UE and simulate the direction and scale of UE.
The research is potentially interesting, giving an additional point of view from one of the countries most concerned by the rapid urbanization process. By the way, although the authors state in several parts of the paper that they analysed a specific case-study in a mountain region, in describing the study area, they do not provide sufficient information on it. Moreover, looking at the methods section, elevation of the study ranges from 147 to 641 meters. I am not sure that this is really a mountain area.
I have to admit that I encountered a lot of trouble in reading this manuscript. This paper need a deep revision in its scientific style. What makes difficult to read it, is the confusion in its style and its content that does not fit the scientific style. The quality of English needs a professional proofing.
Considering that, my revision mostly deals with introduction and materials and methods.
I provided my comments also in the *.pdf file.
Definitely, for this paper, I have to recommend its rejection in its current form. By the way, considering the interesting issue, I suggest to resubmit it. therefore, also following my comments, I suggest the Authors to deeply revise this manuscript prior to re-submit it.
Some important issues to be considered prior the resubmission:
The introduction section contains some of the methodological steps adopted in this paper (i.e., the provided modifications on the original MCR model). I suggest to move these parts on the section 2 (Materials and Methods), leaving just a general problem statement on the most used methodologies to deal with the covered topic.
In
the materials subsection, is not clear what types of data are used and
how the authors managed them in implementing the provided method. In
particular, the authors explain that they interpolated statistical and
monitoring data into the digital raster grid. This does not make sense
if they do not describe the adopted interpolation method and the
geometric resolution. Actually, in the methods subsection, they stated
the geometric resolution of 90x90 m as the optimal. This is very
confusing. Moreover, in the methods subsection, they describe the
selection of resistance factors without having first described the model
and without and explanation of their scientific meaning. In other
words, this manuscript does not provide a general description of the
proposed LMCR model.
On the other hand, while missing important
information, they provided unnecessary and redundant details, such as
the explanation of what is the overlay analysis. On the other hand, they
do not describe how raster data were reclassified in five class (I
mean, why these classes were established).
In the introduction, several methods were introduced (ANN, AHP, etc.) as adopted methods. But, none of them were described in the materials and methods section. For example, I guess that the adopted weights of the so-called resistance factors were obtained from an AHP process following the reclassification using the Saaty’s fundamental scale and the judgments provided by one or more experts. None of this information is provided in the paper.
In the Results section, the Authors describe a Suitability evaluation Index. Further, the direction of Urban expansion and a simulation with different scenario was analysed. Also in this case, no information on the methodological section was provided. On the other hand, also the content on figure 5 should be provided and explained in the methods subsection.
Referring to the English style, this paper has too many sentences that must be revised: plurals, missed words, missed commas, wrong words (formulate instead of formula, etc.). Additionally, some terms are strange and without scientific meaning. Some examples are: ‘ecological land’, ‘ecological environments’.
Specific and technical comments
Lines 34-36: this statement must be supported citing appropriate references.
Line 38: I do not agree with the use of ‘ecological land’ considering that it has no meaning.
Lines 38-39: I suggest to better focus on this statement considering that it represents the focus of this paper. In this direction, additional references should be considered (e.g., https://doi.org/10.5194/esd-3-263-2012; https://link.springer.com/article/10.1007/s10980-008-9253-4; www.tandfonline.com/doi/abs/10.1080/014311600210092; https://esajournals.onlinelibrary.wiley.com/doi/10.1890/1540-9295%282004%29002%5B0249%3ALCBHNA%5D2.0.CO%3B2; https://link.springer.com/chapter/10.1007%2F978-3-642-21928-3_17). On the other, some other concepts on the urban-rural dialectic and on the landscape liveability should be considered (i.e., https://doi.org/10.3390/su10113834).
Lines 68-70: The authors stated that the MCR model is one the best models to assess the ecological connectivity. First, this statement is not supported by any reference. Secondly, this model was proposed in 1992 and, in the last two decades, many researches dealing with this topic have given a strong push forward in design and assess ecological connectivity. Therefore, the authors should justify why they used this model, also giving some reason supported by previous studies.
Lines 81-82: please revise this sentence. Something is missing.
Lines 104-106: The authors explain that they downloaded and used a satellite image with 5.8 of GSD. Therefore, no other metadata on this image was provided.
Author Response
Dear Reviewer
I am very appreciating for your comments and suggestions. My team and I spent 11 days carefully analyzing and revising the structure of the paper (method, result, discussion and conclusion) and especially paid much attention to your comments and suggestions. I would like to re-submit it to ISPRS International Journal of Geo-Information. Title of manuscript has been changed to “Evaluating the suitability of urban expansion based on LMCR model: A case study from Leshan, China” to make it clearer and sooth.
Answers to your questions were as follows:
1. The research is potentially interesting, giving an additional point of view from one of the countries most concerned by the rapid urbanization process. By the way, although the authors state in several parts of the paper that they analyzed a specific case-study in a mountain region, in describing the study area, they do not provide sufficient information on it. Moreover, looking at the methods section, elevation of the study ranges from 147 to 641 meters. I am not sure that this is really a mountain area.
Answer: The absolute height of the study area is really not high, but the terrain is very undulating (147-641), in addition, this scope only refers to the central district and surrounding area, excluding Emei mountain (about 3000m). LS city is defined as a mountain city by the government of LS city, moreover this same statement can also be seen in Sichuan provincial government documents. It may be that LS city is different from those mountain cities, such as those in Europe and America. I am not sure if my explanation is acceptable by you. I took some pictures (see appendix), but the angle is not comprehensive, which maybe help you know about this area.
2. I have to admit that I encountered a lot of trouble in reading this manuscript. This paper need a deep revision in its scientific style. What makes difficult to read it, is the confusion in its style and its content that does not fit the scientific style. The quality of English needs a professional proofing.
Answer: The paper has been revised deeply in its scientific style (see the last version manuscript). In addition, I have got the help of my doctoral supervisor to improve English style.
3. The introduction section contains some of the methodological steps adopted in this paper (i.e., the provided modifications on the original MCR model). I suggest to move these parts on the section 2 (Materials and Methods), leaving just a general problem statement on the most used methodologies to deal with the covered topic.
Answer: I agree with you. The statement of the improved model has been moved to the method section.
4. In the materials subsection, is not clear what types of data are used and how the authors managed them in implementing the provided method. In particular, the authors explain that they interpolated statistical and monitoring data into the digital raster grid. This does not make sense if they do not describe the adopted interpolation method and the geometric resolution. Actually, in the methods subsection, they stated the geometric resolution of 90x90 m as the optimal. This is very confusing.
Answer: Statistical and monitoring data, including the density of urban population (DUP), geological hazards (GH), gross domestic product (GDP) and subgrade bearing capacity (SBC) were interpolated into DRG by Kriging interpolation[25] method, and it’s a geostatistical analysis method, which is suitable for the interpolation of discrete point data. In view of the size of the study area and the spatial resolution of the raw data (DEM), the output raster resolution was set to 30m in buffer analysis and Kriging interpolation. In addition, all DRG were projected into UTM WGS 1984 reference system and clipped with urban bounder. The data preprocessing flow is shown in Figure 5 (See appendix).
The fitting effect of the logic regression model is different in virous scales, therefore, the original DRG data (30m) were re-sampled into five resolutions, specifically, 30m, 90m, 150m, 210m, 270m. Through training and experiment, we found that the logic regression model with 90m resolution passed the relative operating characteristics (ROC) curves test. Therefore, the DRG data of this scale was taken as the most suitable scale for the simulation of urban expansion.
5. Moreover, in the methods subsection, they describe the selection of resistance factors without having first described the model and without and explanation of their scientific meaning. In other words, this manuscript does not provide a general description of the proposed LMCR model.
Answer: the method section, especially the principle of LMCR model, I consulted the literature and discussed many times with my team, and revised the statement of the LMCR model strictly.
We divided a heterogeneous landscape into a “source” or “sink” landscape according to eco-process theory, of which the source landscape refers to those landscape that can promote the development of the process, and "sink" landscape refers to those landscape that can prevent or delay the development of the process. The qualities of "source" and "sink" landscape is relative, a "source" landscape in one process, it may be "sink" landscape in another process, accordingly, the analysis of "source" and "sink" landscape must be directed at a specific process. The source and sink landscape theory elaborated is to explore dynamic balance relationship between urban expansion and ecological protection, which in turn realize a suitable spatial pattern for urban expansion. The transformation between source and sink is a competitive process, which is performed through overcoming various resistance forces. And then all the resistance forces are integrated into a logic resistance surface. LMCR model essentially reflects the degree of resistance of urban expansion sources to sinks in the process of expansion. Of course, the degree of resistance varies with different sinks, which can be achieved by exploring the logical relationship between expansion sources and sinks. In addition, the cost distance in the model is not the actual distance, but the reflection of the spatial relationship between the two landscape units. We achieved the spatial relationship by calculating the drag coefficient when the expansion source passes through other landscape units. We can determine the connectivity between the two units based on the value of the minimum cumulative resistance, when the urban expansion source passes through a specific landscape, and usually the minimum cumulative resistance path is the suitable path of urban expansion.
6. On the other hand, while missing important information, they provided unnecessary and redundant details, such as the explanation of what is the overlay analysis. On the other hand, they do not describe how raster data were reclassified in five class (I mean, why these classes were established).
Answer: The grid values and units of the 10 resistance factors are different, such as elevation data, which ranges from 147 to 641 in meters, while other resistance factors (SLOPE (°), GH (sq. km), SBC (t/m2), LST, NR, DUP (sq. km), CT (km), WSDC (km), GDP ()) are different, especially LST and NR have no the data units. Therefore, it is meaningless to calculate these 10 resistance factors by overlay analysis. To solve this problem, we need to reclassify and assign the resistance factor data (see table 2), and convert the data into comparable dimensionless values. There are five types of landscape in LST, accordingly we also reclassify other resistance factors into five categories to avoid null values due to excessive or insufficient categories. Moreover, the principle of assignment is the simple data structure and easy to calculate, because this assignment is dimensionless and will not affect the calculation results. The overlay analysis of a GIS calculation method was performed based on the resistance factors assigned and its weight to calculate the logic resistance surface.
7. In the introduction, several methods were introduced (ANN, AHP, etc.) as adopted methods. But, none of them were described in the materials and methods section. For example, I guess that the adopted weights of the so-called resistance factors were obtained from an AHP process following the reclassification using the Saaty’s fundamental scale and the judgments provided by one or more experts. None of this information is provided in the paper.
Answer: In the 2.3.3.2.2 section, we explored the relationship between the expansion source and resistance factors and got the logic regression coefficient. The coefficient is particularly vital because it is the weight of the logic resistance surface that we calculate in the following process. It should be noted that this weight is different from the weight obtained through AHP method, which is based on the experiences or the judgments provided by one or more experts.
8. In the Results section, the Authors describe a Suitability evaluation Index. Further, the direction of Urban expansion and a simulation with different scenario was analysed. Also in this case, no information on the methodological section was provided. On the other hand, also the content on figure 5 should be provided and explained in the methods subsection.
Answer: We have revised the method section strictly. Specifically, you can view the method section in the new manuscript
Based on the LMCR surface, three aspects of analysis can be performed to answer the three questions posed in the introduction section. (1) We extracted the section line of LMCR surface from UGC to satellite town as the resistance curve to evaluate the resistance variation in the different direction. (2) Evaluating the suitability of urban expansion, we used the LMCR surface to conduct the suitability zoning (reclassify method), which in turn achieve the suitability distribution of urban expansion. (3) The expansion path and direction were extracted from the LMCR surface by the cost path analysis. Moreover, we combine the urban planning of LS for 2020 and 2030, the urban scale will be controlled at 100 km2 and 140 km2 to balance the construction land and ecological space and realize the sustainable urban development. In view of this, four scenarios were designed, namely, 80km2, 100km2, 140km2 and 160km2, to analyze the connectivity of urban central areas and satellite towns and the protection of urban ecological space. In detail, the expansion source and ecological barrier were used as input data for buffer analysis, and buffer scale of four scenarios was set.
9. Referring to the English style, this paper has too many sentences that must be revised: plurals, missed words, missed commas, wrong words (formulate instead of formula, etc.). Additionally, some terms are strange and without scientific meaning. Some examples are: ‘ecological land’, ‘ecological environments’.
Answer: This English style has been revised by word for word and I have got the help of doctoral supervisor. In addition, for the revision of special vocabulary, we read some books and papers, especially you recommended papers.
Specific and technical comments
10. Lines 34-36: this statement must be supported citing appropriate references.
Answer: I have cited appropriate references to support this statement in the introduction section and the specific references id is from [1] to [3].
11. Line 38: I do not agree with the use of ‘ecological land’ considering that it has no meaning.
Answer: The vocabulary has been revised in the new manuscript.
12. Lines 38-39: I suggest to better focus on this statement considering that it represents the focus of this paper. In this direction, additional references should be considered (e.g., https://doi.org/10.5194/esd-3-263-2012; https://link.springer.com/article/10.1007/s10980-008-9253-4;www.tandfonline.com/doi/abs/10.1080/014311600210092; https://esajournals.onlinelibrary.wiley.com/doi/10.1890/1540-9295%282004%29002%5B0249%3ALCBHNA%5D2.0.CO%3B2; https://link.springer.com/chapter/10.1007%2F978-3-642-21928-3_17). On the other, some other concepts on the urban-rural dialectic and on the landscape live ability should be considered (i.e., https://doi.org/10.3390/su10113834).
Answer: I have carefully read and analyzed these articles you recommended to improve my manuscript. Thank you so much….
13. Lines 68-70: The authors stated that the MCR model is one the best models to assess the ecological connectivity. First, this statement is not supported by any reference. Secondly, this model was proposed in 1992 and, in the last two decades, many researches dealing with this topic have given a strong push forward in design and assess ecological connectivity. Therefore, the authors should justify why they used this model, also giving some reason supported by previous studies.
Answer: The statement about "one of the best methods" isn’t very accurate, therefore the it was revised to “an effective method and cited appropriate references to support it.
14. Lines 81-82: please revise this sentence. Something is missing.
Answer: yes, the sentence has been revised.
15. Lines 104-106: The authors explain that they downloaded and used a satellite image with 5.8 of GSD. Therefore, no other metadata on this image was provided.
Answer: Resources No.3 (ZY03) satellite remote sensing image (July 2018) were downloaded from China Centre for Resources Satellite Data and Applications (CCRSDA). The image was taken by a multi-spectral camera mounted on ZY03 satellite. It has four spectral bands with a wavelength range of 0.45-0.89μm, a spatial resolution of 5.8m and the width of 51km (see fig.4)

Round 2
Reviewer 1 Report
The paper is obviously improved after revision. Just some problems need to be treated.
1. The abstract is still not so good. The significance and innovation of the paper is not clearly described. If possible, we suggest the authors to revise the abstract again for better expression. Why is this research important? What is the main difference between mountain city and other cities? How the authors treated the problem? How effective was the method?
2. In line 38, when the reference is a source of statistics (not a academic research), the sentence with citation in the main text should be a concrete description of the digits in the source of statistics, rather than an expression of opinion.
3. In line 212, line 233, line 249, the number of hierarchical levels of subtitles should be exceed 4. For most of time, 3 levels of subtitle is enough.
Reviewer 2 Report
The quality of the paper has been improved. But I still have 2 questions.
Q1 : Equation 1 is still causing indices problems : i starts by 1 and what is the upper limit, the same problem with j ?
Q2 : In equation 2, I still do not understand the addition operations, in the paper the authors said that the logic regression is used, thus the operator should be logic operators ?
Reviewer 4 Report
Dear authors,
I read this revised version of your paper. Actually, you improved your paper but you had should take more attention in doing this revised version.
Therefore, I still have several concerns, especially with the description of the methods. Additionally, part of my previous comments were not considered by the authors.
Also referring to the added parts, several sentences that describe the methodological aspects are not acceptable in a paper published in a high quality scientific journal. In the following examples I reported some sentences that whether do not have any sense or are wrong:
Line 82: “It would be a meaningful exploration to realize the sustainable development of mountain”.
Lines 132-134: “The discrete data were converted to the continuous digital raster graphic (DRG) by buffer analyst (analysis), such as roads and rivers were transformed to the distance to roads and rivers, namely, the convenient transportation (CT) and the water supply-drainage conditions (WSDC)”.
Lines 136-138: “[….] were interpolated into DRG by Kriging interpolation [30] method, and it’s a geostatistical analysis method, which is suitable for the interpolation of discrete point data”.
Definitely, to give a chance to the authors I agree for a major revision of the present manuscript. By the way, what I suggest is to provide a deep review of their manuscript that need a lot of work. In doing that, I suggest the authors to follow all reviewers’ comments and revise the manuscript following the scientific style.